# Algorithmic recourse under imperfect causal knowledge: a probabilistic approach

**Amir-Hossein Karimi**[*,1,2]   **Julius von Kügelgen**[*,1,3]   **Bernhard Schölkopf** [1]   **Isabel Valera** [1,4]

[1]Max Planck Institute for Intelligent Systems, Tübingen, Germany
[2]Max Planck ETH Center for Learning Systems, Zürich, Switzerland
[3]Department of Engineering, University of Cambridge, United Kingdom
[4]Department of Computer Science, Saarland University, Saarbrücken, Germany
`{amir, jvk, bs, ivalera}@tue.mpg.de`

## Abstract

Recent work has discussed the limitations of counterfactual explanations to recommend actions for algorithmic recourse, and argued for the need of taking causal relationships between features into consideration. Unfortunately, in practice, the true underlying structural causal model is generally unknown. In this work, we first show that it is impossible to guarantee recourse without access to the true structural equations. To address this limitation, we propose two probabilistic approaches to select optimal actions that achieve recourse with high probability given limited causal knowledge (e.g., only the causal graph). The first captures uncertainty over structural equations under additive Gaussian noise, and uses Bayesian model averaging to estimate the counterfactual distribution. The second removes any assumptions on the structural equations by instead computing the average effect of recourse actions on individuals similar to the person who seeks recourse, leading to a novel subpopulation-based interventional notion of recourse. We then derive a gradient-based procedure for selecting optimal recourse actions, and empirically show that the proposed approaches lead to more reliable recommendations under imperfect causal knowledge than non-probabilistic baselines.

## 1   Introduction

As machine learning algorithms are increasingly used to assist consequential decision making in a wide range of real-world settings [36, 41], providing explanations for the decision of these black-box models becomes crucial [7, 58]. A popular approach is that of (nearest) *counterfactual explanations*, which refer to the closest feature instantiations that would have resulted in a changed prediction [59]. While providing some insight (explanation) into the underlying black-box classifier, such counterfactual explanations do not directly translate into actionable recommendations to individuals for obtaining a more favourable prediction[22, 5]—a related task referred to as *algorithmic recourse* [54, 55, 19, 21]. Importantly, prior work on both counterfactual explanations and algorithmic recourse treats features as independently manipulable inputs, thus *ignoring the causal relationships between features*.

In this context, recent work [22] has argued for the need of taking into account the causal structure between features to find a minimal set of actions (in the form of interventions) that guarantees recourse. However, while this approach is theoretically sound, it involves computing counterfactuals in the *true underlying structural causal model* (SCM) [35], and thus relies on strong impractical

---

[*]Equal contribution

assumptions; specifically, it requires complete knowledge of the true structural equations. While for many applications it is possible to draw a causal diagram from expert knowledge, assumptions about the form of structural equations are, in general, not testable and may thus not hold in practice [38]. As a result, counterfactuals computed using a misspecified causal model may be inaccurate and recommend actions that are sub-optimal or, even worse, ineffective to achieve recourse.

In this work, we focus on the problem of algorithmic recourse when only *limited causal knowledge* is available (as it is generally the case). To this end, we propose two probabilistic approaches which allow to relax the strong assumption of a fully-specified SCM made in [22]. In the first approach, we assume that, while the underlying SCM is unknown, it belongs to the family of additive Gaussian noise models [16, 37]. We then make use of Gaussian processes (GPs) [62] to average predictions over a whole family of SCMs and thus to obtain a distribution over *counterfactual* outcomes which forms the basis for *individualised* algorithmic recourse. The second approach considers a different *subpopulation-based* notion of algorithmic recourse by estimating the effect of *interventions* for individuals similar to the one for which we aim to achieve recourse. It thus addresses a different (rung 2) target quantity than the counterfactual/individualised (rung 3) approach which allows us to further relax our assumptions by removing any assumptions on the form of the structural equations. This approach is based on the idea of the conditional average treatment effect (CATE) [1], and relies on conditional variational autoencoders (CVAEs) [48] to estimate the interventional distribution. In both cases, we assume that the causal graph is known or can be postulated from expert knowledge, as without such an assumption causal reasoning from observational data is not possible [38, Prop. 4.1].

In more detail, we first demonstrate as a motivating negative result that recourse guarantees are only possible if the true SCM is known (§3). Then, we introduce two probabilistic approaches for handling different levels of uncertainty in the structural equations (§4 and §5), and propose a gradient-based method to find a set of actions that achieves recourse with a given probability at minimum cost (§6). Our experiments (§7) on synthetic and semi-synthetic loan approval data, show the need for probabilistic approaches to achieve algorithmic recourse in practice, as point estimates of the underlying true SCM often propose invalid recommendations or achieve recourse only at higher cost. Importantly, our results also show that subpopulation-based recourse is the right approach to adopt when assumptions such as additive noise do not hold. A user-friendly implementation of all methods that only requires specification of the causal graph and a training set is available at https://github.com/amirhk/recourse.

## 2   Background and related work

**Causality: structural causal models, interventions, and counterfactuals.**   To reason formally about causal relations between features $\mathbf{X} = \{X_1, ..., X_d\}$, we adopt the *structural causal model* (SCM) framework [35].[2] Specifically, we assume that the data-generating process of $\mathbf{X}$ is described by an (unknown) underlying SCM $\mathcal{M}$ of the general form

$$\mathcal{M} = (\mathbf{S}, P_{\mathbf{U}}), \quad \mathbf{S} = \{X_r := f_r(\mathbf{X}_{\mathrm{pa}(r)}, U_r)\}_{r=1}^d, \quad P_{\mathbf{U}} = P_{U_1} \times \ldots \times P_{U_d}, \tag{1}$$

where the structural equations $\mathbf{S}$ are a set of assignments generating each observed variable $X_r$ as a deterministic function $f_r$ of its causal parents $\mathbf{X}_{\mathrm{pa}(r)} \subseteq \mathbf{X} \setminus X_r$ and an unobserved noise variable $U_r$. The assumption of mutually independent noises (i.e., a fully factorised $P_{\mathbf{U}}$) entails that there is no hidden confounding and is referred to as *causal sufficiency*. An SCM is often illustrated by its associated causal graph $\mathcal{G}$, which is obtained by drawing a directed edge from each node in $\mathbf{X}_{\mathrm{pa}(r)}$ to $X_r$ for $r \in [d] := \{1, \ldots, d\}$, see Fig. 1b and 1c for an example. We assume throughout that $\mathcal{G}$ is acyclic. In this case, $\mathcal{M}$ implies a unique observational distribution $P_{\mathbf{X}}$, which factorises over $\mathcal{G}$, defined as the push-forward of $P_{\mathbf{U}}$ via $\mathbf{S}$.[3]

Importantly, the SCM framework also entails *interventional distributions* describing a situation in which some variables are manipulated externally. E.g., using the *do*-operator, an intervention which fixes $\mathbf{X}_{\mathcal{I}}$ to $\boldsymbol{\theta}$ (where $\mathcal{I} \subseteq [d]$) is denoted by $do(\mathbf{X}_{\mathcal{I}} = \boldsymbol{\theta})$. The corresponding distribution of the remaining variables $\mathbf{X}_{-\mathcal{I}}$ can be computed by replacing the structural equations for $\mathbf{X}_{\mathcal{I}}$ in $\mathbf{S}$ to obtain

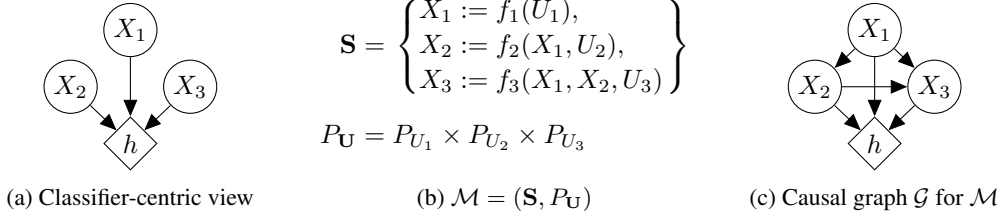

(a) Classifier-centric view      (b) $\mathcal{M} = (\mathbf{S}, P_{\mathbf{U}})$      (c) Causal graph $\mathcal{G}$ for $\mathcal{M}$

Figure 1: A view commonly adopted for counterfactual explanations (a) treats features as independently manipulable inputs to a given fixed and deterministic classifier $h$. In the causal approach to algorithmic recourse taken in this work, we instead view variables as causally related to each other by a structural causal model (SCM) $\mathcal{M}$ (b) with associated causal graph $\mathcal{G}$ (c).

the new set of equations $\mathbf{S}^{do(\mathbf{X}_{\mathcal{I}}=\boldsymbol{\theta})}$. The interventional distribution $P_{\mathbf{X}_{-\mathcal{I}}|do(\mathbf{X}_{\mathcal{I}}=\boldsymbol{\theta})}$ is then given by the observational distribution implied by the manipulated SCM $\left(\mathbf{S}^{do(\mathbf{X}_{\mathcal{I}}=\boldsymbol{\theta})}, P_{\mathbf{U}}\right)$.

Similarly, an SCM also implies distributions over *counterfactuals*—statements about a world in which a hypothetical intervention was performed *all else being equal*. For example, *given* observation $\mathbf{x}^{\mathrm{F}}$ we can ask what would have happened if $\mathbf{X}_{\mathcal{I}}$ had instead taken the value $\boldsymbol{\theta}$. We denote the counterfactual variable by $\mathbf{X}(do(\mathbf{X}_{\mathcal{I}} = \boldsymbol{\theta}))|\mathbf{x}^{\mathrm{F}}$, whose distribution can be computed in three steps [35]:

*1. Abduction*: compute the posterior distribution over background variables given $\mathbf{x}^{\mathrm{F}}$, $P_{\mathbf{U}|\mathbf{x}^{\mathrm{F}}}$;

*2. Action:* perform the intervention to obtain the new structural equations $\mathbf{S}^{do(\mathbf{X}_{\mathcal{I}}=\boldsymbol{\theta})}$; and,

*3. Prediction:* $P_{\mathbf{X}(do(\mathbf{X}_{\mathcal{I}}=\boldsymbol{\theta}))|\mathbf{x}^{\mathrm{F}}}$ is the distribution induced by the resulting SCM $\left(\mathbf{S}^{do(\mathbf{X}_{\mathcal{I}}=\boldsymbol{\theta})}, P_{\mathbf{U}|\mathbf{x}^{\mathrm{F}}}\right)$.

**Explainable ML: "counterfactual" explanations and (causal) algorithmic recourse.** Assume that we are given a binary probabilistic classifier $h : \mathcal{X} \to [0, 1]$ trained to make decisions about i.i.d. samples from the data distribution $P_{\mathbf{X}}$.[4] For ease of illustration, we adopt the setting of loan approval as a running example, i.e., $h(\mathbf{x}) \geq 0.5$ denotes that a loan is granted and $h(\mathbf{x}) < 0.5$ that it is denied. For a given individual $\mathbf{x}^{\mathrm{F}}$ that was denied a loan, $h(\mathbf{x}^{\mathrm{F}}) < 0.5$, we aim to answer the following questions: "Why did individual $\mathbf{x}^{\mathrm{F}}$ not get the loan?" and "What would they have to change, preferably with minimal effort, to increase their chances for a future application?".

A popular approach to this task is to find so-called (nearest) *counterfactual explanations* [59], where the term "counterfactual" is meant in the sense of the closest possible world with a different outcome [30]. Translating this idea to our setting, a counterfactual explanation $\mathbf{x}^{\mathrm{CE}}$ for an individual $\mathbf{x}^{\mathrm{F}}$ is given by a solution to the following optimisation problem:

$$\mathbf{x}^{\mathrm{CE}} \in \arg\min_{\mathbf{x}\in\mathcal{X}} \quad \mathrm{dist}(\mathbf{x}, \mathbf{x}^{\mathrm{F}}) \quad \text{subject to} \quad h(\mathbf{x}) \geq 0.5, \tag{2}$$

where $\mathrm{dist}(\cdot, \cdot)$ is a similarity metric on $\mathcal{X}$, and additional constraints may be added to reflect plausibility, feasibility, or diversity of the obtained counterfactual explanations [19, 20, 32, 33, 39, 44].

Importantly, while $\mathbf{x}^{\mathrm{CE}}$ signifies the most similar individual to $\mathbf{x}^{\mathrm{F}}$ that would receive the loan, it does not inform $\mathbf{x}^{\mathrm{F}}$ on the actions they should perform to become $\mathbf{x}^{\mathrm{CE}}$. To address this limitation, the recently proposed framework of *algorithmic recourse* focuses instead on the actions an individual can perform to achieve a more favourable outcome [54]. The emphasis is thus shifted from minimising a distance as in (2) to optimising a personalised cost function $\mathrm{cost}^{\mathrm{F}}(\cdot)$ over a set of actions $\mathbb{A}^{\mathrm{F}}$ which individual $\mathbf{x}^{\mathrm{F}}$ can perform. However, most prior work on both counterfactual explanations and algorithmic recourse considers features as independently manipulable inputs to the classifier $h$ (see Fig. 1a), and therefore, ignores the potentially rich causal structure over $\mathbf{X}$ (see Fig. 1c). A number of authors have argued for the need to consider causal relations between variables when generative counterfactual explanations [59, 54, 20, 33, 32], however, the resulting counterfactuals fail to imply feasible and optimal recourse actions [22].

In the most relevant work to the current [22], the authors approach the algorithmic recourse problem from a causal perspective within the SCM framework and propose to view recourse actions $a \in \mathbb{A}^{\mathrm{F}}$ as interventions of the form $do(\mathbf{X}_{\mathcal{I}} = \boldsymbol{\theta})$. For the class of invertible SCMs, such as additive noise

models (ANM) [16], where the structural equations $\mathbf{S}$ are of the form

$$\mathbf{S} = \{X_r := f_r(\mathbf{X}_{\mathrm{pa}(r)}) + U_r\}_{r=1}^d \implies u_r^{\mathrm{F}} = x_r^{\mathrm{F}} - f_r(\mathbf{x}_{\mathrm{pa}(r)}^{\mathrm{F}}), \quad r \in [d], \tag{3}$$

they propose to use the three steps of structural counterfactuals in [35] to assign a single counterfactual $\mathbf{x}^{\mathrm{SCF}}(a) := \mathbf{x}(a)|\mathbf{x}^{\mathrm{F}}$ to each action $a = do(\mathbf{X}_{\mathcal{I}} = \boldsymbol{\theta}) \in \mathbb{A}^{\mathrm{F}}$, and solve the optimisation problem,

$$a^{\mathrm{F}} = \arg\min_{a=do(\mathbf{X}_{\mathcal{I}}=\boldsymbol{\theta})\in\mathbb{A}^{\mathrm{F}}} \mathrm{cost}^{\mathrm{F}}(a) \quad \text{subject to} \quad h(\mathbf{x}^{\mathrm{SCF}}(a)) \geq 0.5. \tag{4}$$

## 3 Negative result: no recourse guarantees for unknown structural equations

In practice, the structural counterfactual $\mathbf{x}^{\mathrm{SCF}}(a)$ can only be computed using an approximate (and likely imperfect) SCM $\mathcal{M} = (\mathbf{S}, P_{\mathbf{U}})$, which is estimated from data assuming a particular form of the structural equation as in (3). However, assumptions on the form of $\mathbf{S}_\star$ are generally untestable—not even with a randomised experiment—since there exist multiple SCMs which imply the same observational and interventional distributions, but entail different structural counterfactuals.

**Example 1** (adapted from 6.19 in [38]). *Consider the following two SCMs $\mathcal{M}_A$ and $\mathcal{M}_B$ which arise from the general form in Figure 1b by choosing $U_1, U_2 \sim Bernoulli(0.5)$ and $U_3 \sim Uniform(\{0, \ldots, K\})$ independently in both $\mathcal{M}_A$ and $\mathcal{M}_B$, with structural equations*

$$\begin{aligned} X_1 &:= U_1, & in \quad &\{\mathcal{M}_A, \mathcal{M}_B\}, \\ X_2 &:= X_1(1 - U_2), & in \quad &\{\mathcal{M}_A, \mathcal{M}_B\}, \\ X_3 &:= \mathbb{I}_{X_1 \neq X_2}(\mathbb{I}_{U_3>0}X_1 + \mathbb{I}_{U_3=0}X_2) + \mathbb{I}_{X_1=X_2}U_3, & in \quad &\mathcal{M}_A, \\ X_3 &:= \mathbb{I}_{X_1 \neq X_2}(\mathbb{I}_{U_3>0}X_1 + \mathbb{I}_{U_3=0}X_2) + \mathbb{I}_{X_1=X_2}(K - U_3), & in \quad &\mathcal{M}_B. \end{aligned}$$

*Then $\mathcal{M}_A$ and $\mathcal{M}_B$ both imply exactly the same observational and interventional distributions, and thus are indistinguishable from empirical data. However, having observed $\mathbf{x}^{\mathrm{F}} = (1, 0, 0)$, they predict different counterfactuals had $X_1$ been 0, i.e., $\mathbf{x}^{\mathrm{SCF}}(X_1 = 0) = (0, 0, 0)$ and $(0, 0, K)$, respectively.[5]*

Confirming or refuting an assumed form of $\mathbf{S}_\star$ would thus require counterfactual data which is, by definition, never available. Thus, example 1 proves the following proposition by contradiction.

**Proposition 2** (Lack of recourse guarantees). *Unless the set of descendants of intervened-upon variables is empty, algorithmic recourse can, in general, be guaranteed only if the true structural equations are known, irrespective of the amount and type of available data.*

**Remark 3.** *The converse of Proposition 2 does not hold. E.g., given $\mathbf{x}^{\mathrm{F}} = (1, 0, 1)$ in Example 1, abduction in either model yields $U_3 > 0$, so the counterfactual of $X_3$ cannot be predicted exactly.*

Building on the framework in [22], we next present two novel approaches for causal algorithmic recourse under unknown structural equations. The first approach in §4 aims to estimate the counterfactual distribution under the assumption of ANMs (3) with Gaussian noise for the structural equations. The second approach in §5 makes no assumptions about the structural equations, and instead of approximating the structural equations, it considers the effect of interventions on a sub-population similar to $\mathbf{x}^{\mathrm{F}}$. We recall that the causal graph is assumed to be known throughout.

## 4 Individualised algorithmic recourse via (probabilistic) counterfactuals

Since the true SCM $\mathcal{M}_\star$ is unknown, one approach to solving (4) is to learn an approximate SCM $\mathcal{M}$ within a given model class from training data $\{\mathbf{x}^i\}_{i=1}^n$. For example, for an ANM (3) with zero-mean noise, the functions $f_r$ can be learned via linear or kernel (ridge) regression of $X_r$ given $\mathbf{X}_{\mathrm{pa}(r)}$ as input. We refer to these approaches as $\mathcal{M}_{\mathrm{LIN}}$ and $\mathcal{M}_{\mathrm{KR}}$, respectively. $\mathcal{M}$ can then be used in place of $\mathcal{M}_\star$ to infer the noise values as in (3), and subsequently to predict a *single-point counterfactual* $\mathbf{x}^{\mathrm{SCF}}(a)$ to be used in (4). However, the learned causal model $\mathcal{M}$ may be imperfect, and thus lead to wrong counterfactuals due to, e.g., the finite sample of the observed data, or more importantly, due to model misspecification (i.e., assuming a wrong parametric form for the structural equations).

To solve such limitation, we adopt a Bayesian approach to account for the uncertainty in the estimation of the structural equations. Specifically, we assume additive Gaussian noise and rely on probabilistic regression using a Gaussian process (GP) prior over the functions $f_r$ [62].

**Definition 4** (GP-SCM). *A Gaussian process* SCM *(GP-SCM) over* $\mathbf{X}$ *refers to the model*

$$X_r := f_r(\mathbf{X}_{pa(r)}) + U_r, \qquad f_r \sim \mathcal{GP}(0, k_r), \qquad U_r \sim \mathcal{N}(0, \sigma_r^2), \qquad r \in [d], \qquad (5)$$

*with covariance functions* $k_r : \mathcal{X}_{pa(r)} \times \mathcal{X}_{pa(r)} \to \mathbb{R}$, *e.g.,* RBF *kernels for continuous* $X_{pa(r)}$.

While GPs have previously been studied in a causal context for structure learning [13, 56], estimating treatment effects [2, 43], or learning SCMs with latent variables and measurement error [47], our goal here is to account for the uncertainty over $f_r$ in the computation of the posterior over $U_r$, and thus to obtain a *counterfactual distribution*, as summarised in the following propositions.

**Proposition 5** (GP-SCM noise posterior). *Let* $\{\mathbf{x}^i\}_{i=1}^n$ *be an observational sample from* (5). *For each* $r \in [d]$ *with non empty parent set* $|pa(r)| > 0$, *the posterior distribution of the noise vector* $\mathbf{u}_r = (u_r^1, ..., u_r^n)$, *conditioned on* $\mathbf{x}_r = (x_r^1, ..., x_r^n)$ *and* $\mathbf{X}_{pa(r)} = (\mathbf{x}_{pa(r)}^1, ..., \mathbf{x}_{pa(r)}^n)$, *is given by*

$$\mathbf{u}_r | \mathbf{X}_{pa(r)}, \mathbf{x}_r \sim \mathcal{N}\left(\sigma_r^2 (\mathbf{K} + \sigma_r^2 \mathbf{I})^{-1} \mathbf{x}_r, \sigma_r^2 \left(\mathbf{I} - \sigma_r^2 (\mathbf{K} + \sigma_r^2 \mathbf{I})^{-1}\right)\right), \qquad (6)$$

*where* $\mathbf{K} := \left(k_r\left(\mathbf{x}_{pa(r)}^i, \mathbf{x}_{pa(r)}^j\right)\right)_{ij}$ *denotes the Gram matrix.*

Next, in order to compute counterfactual distributions, we rely on ancestral sampling (according to the causal graph) of the descendants of the intervention targets $\mathbf{X}_{\mathcal{I}}$ using the noise posterior of (6). The counterfactual distribution of each descendant $X_r$ is given by the following proposition.

**Proposition 6** (GP-SCM counterfactual distribution). *Let* $\{\mathbf{x}^i\}_{i=1}^n$ *be an observational sample from* (5). *Then, for* $r \in [d]$ *with* $|pa(r)| > 0$, *the counterfactual distribution over* $X_r$ *had* $\mathbf{X}_{pa(r)}$ *been* $\tilde{\mathbf{x}}_{pa(r)}$ *(instead of* $\mathbf{x}_{pa(r)}^F$) *for individual* $\mathbf{x}^F \in \{\mathbf{x}^i\}_{i=1}^n$ *is given by*

$$X_r(\mathbf{X}_{pa(r)} = \tilde{\mathbf{x}}_{pa(r)}) | \mathbf{x}^F, \{\mathbf{x}^i\}_{i=1}^n \sim \mathcal{N}\left(\mu_r^F + \tilde{\mathbf{k}}^T (\mathbf{K} + \sigma_r^2 \mathbf{I})^{-1} \mathbf{x}_r, s_r^F + \tilde{k} - \tilde{\mathbf{k}}^T (\mathbf{K} + \sigma_r^2 \mathbf{I})^{-1} \tilde{\mathbf{k}}\right), \quad (7)$$

*where* $\tilde{k} := k_r(\tilde{\mathbf{x}}_{pa(r)}, \tilde{\mathbf{x}}_{pa(r)})$, $\tilde{\mathbf{k}} := \left(k_r(\tilde{\mathbf{x}}_{pa(r)}, \mathbf{x}_{pa(r)}^1), \ldots, k_r(\tilde{\mathbf{x}}_{pa(r)}, \mathbf{x}_{pa(r)}^n)\right)$, $\mathbf{x}_r$ *and* $\mathbf{K}$ *as defined in Proposition 5, and* $\mu_r^F$ *and* $s_r^F$ *are the posterior mean and variance of* $u_r^F$ *given by* (6).

All proofs can be found in Appendix A. We can now generalise the recourse problem (4) to our probabilistic setting by replacing the single-point counterfactual $\mathbf{x}^{\text{SCF}}(a)$ with the counterfactual random variable $\mathbf{X}^{\text{SCF}}(a) := \mathbf{X}(a) | \mathbf{x}^F$. As a consequence, it no longer makes sense to consider a hard constraint of the form $h(\mathbf{x}^{\text{SCF}}(a)) > 0.5$, i.e., that the prediction needs to change. Instead, we can reason about the expected classifier output under the counterfactual distribution, leading to the following *probabilistic version of the individualised recourse optimisation problem*:

$$\min_{a = do(\mathbf{X}_{\mathcal{I}} = \boldsymbol{\theta}) \in \mathbb{A}^F} \text{cost}^F(a) \quad \text{subject to} \quad \mathbb{E}_{\mathbf{X}^{\text{SCF}}(a)}\left[h\left(\mathbf{X}^{\text{SCF}}(a)\right)\right] \geq \texttt{thresh}(a). \qquad (8)$$

Note that the threshold $\texttt{thresh}(a)$ is allowed to depend on $a$. For example, an intuitive choice is

$$\texttt{thresh}(a) = 0.5 + \gamma_{\text{LCB}} \sqrt{\text{Var}_{\mathbf{X}^{\text{SCF}}(a)}\left[h\left(\mathbf{X}^{\text{SCF}}(a)\right)\right]} \qquad (9)$$

which has the interpretation of the lower-confidence bound crossing the decision boundary of $0.5$. Note that larger values of the hyperparameter $\gamma_{\text{LCB}}$ lead to a more conservative approach to recourse, while for $\gamma_{\text{LCB}} = 0$ merely crossing the decision boundary with $\geq 50\%$ chance suffices.

## 5 Subpopulation-based algorithmic recourse via interventions and CATEs

The GP-SCM approach in §4 allows us to average over an infinite number of (non-)linear structural equations, under the assumption of additive Gaussian noise. However, this assumption may still not hold under the true SCM, leading to sub-optimal or inefficient solutions to the recourse problem. Next, we remove any assumptions about the structural equations, and propose a second approach that does not aim to approximate an individualised counterfactual distribution, but instead considers the effect of interventions on a subpopulation defined by certain shared characteristics with the given (factual) individual $\mathbf{x}^F$. The key idea behind this approach resembles the notion of conditional average treatment effects (CATE) [1] (illustrated in Fig. 2a) and is based on the fact that any intervention $do(\mathbf{X}_{\mathcal{I}} = \boldsymbol{\theta})$ only influences the descendants $\text{d}(\mathcal{I})$ of the intervened-upon variables, while the non-descendants $\text{nd}(\mathcal{I})$ remain unaffected. Thus, when evaluating an intervention, we can condition on $\mathbf{X}_{\text{nd}(\mathcal{I})} = \mathbf{x}_{\text{nd}(\mathcal{I})}^F$, thus selecting a subpopulation of individuals similar to the factual subject.

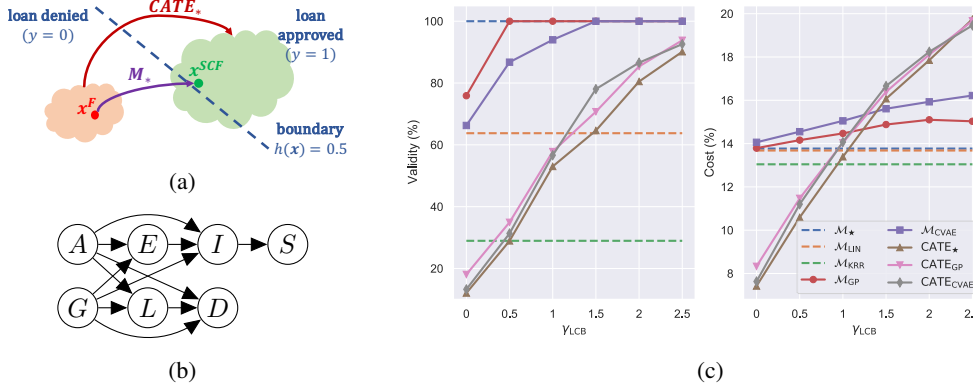

Figure 2: (a) Illustration of point- and subpopulation-based recourse approaches. (b) Assumed causal graph for the semi-synthetic loan approval dataset. (c) Trade-off between validity and cost which can be controlled via $\gamma_{\text{LCB}}$ for the probabilistic recourse methods.

Specifically, we propose to solve the following *subpopulation-based recourse optimisation problem*

$$\min_{a \in \mathbb{A}^{\text{F}}} \quad \text{cost}^{\text{F}}(a) \quad \text{subject to} \quad \mathbb{E}_{\mathbf{X}_{\text{d}(\mathcal{I})}|do(\mathbf{X}_{\mathcal{I}}=\boldsymbol{\theta}),\mathbf{x}^{\text{F}}_{\text{nd}(\mathcal{I})}} \big[ h\big(\mathbf{x}^{\text{F}}_{\text{nd}(\mathcal{I})}, \boldsymbol{\theta}, \mathbf{X}_{\text{d}(\mathcal{I})}\big) \big] \geq \texttt{thresh}(a), \quad (10)$$

where, in contrast to (8), the expectation is taken over the corresponding interventional distribution.

In general, this interventional distribution does not match the conditional distribution, i.e., $P_{\mathbf{X}_{\text{d}(\mathcal{I})}|do(\mathbf{X}_{\mathcal{I}}=\boldsymbol{\theta}),\mathbf{x}^{\text{F}}_{\text{nd}(\mathcal{I})}} \neq P_{\mathbf{X}_{\text{d}(\mathcal{I})}|\mathbf{X}_{\mathcal{I}}=\boldsymbol{\theta},\mathbf{x}^{\text{F}}_{\text{nd}(\mathcal{I})}}$, because some spurious correlations in the observational distribution do not transfer to the interventional setting. For example, in Fig. 1c we have that $P_{X_2|do(X_1=x_1,X_3=x_3)} = P_{X_2|X_1=x_1} \neq P_{X_2|X_1=x_1,X_3=x_3}$. Fortunately, the interventional distribution can still be identified from the observational one, as stated in the following proposition.

**Proposition 7.** *Subject to causal sufficiency, $P_{\mathbf{X}_{d(\mathcal{I})}|do(\mathbf{X}_{\mathcal{I}}=\boldsymbol{\theta}),\mathbf{x}^{F}_{nd(\mathcal{I})}}$ is observationally identifiable:*

$$p\big(\mathbf{X}_{d(\mathcal{I})}|do(\mathbf{X}_{\mathcal{I}} = \boldsymbol{\theta}), \mathbf{x}^{F}_{nd(\mathcal{I})}\big) = \prod_{r \in d(\mathcal{I})} p\big(X_r | \mathbf{X}_{pa(r)}\big)\Big|_{\mathbf{X}_{\mathcal{I}}=\boldsymbol{\theta}, \mathbf{X}_{nd(\mathcal{I})}=\mathbf{x}^{F}_{nd(\mathcal{I})}}. \quad (11)$$

As evident from Proposition 7, tackling the optimisation problem in (10) in the general case (i.e., for arbitrary graphs and intervention sets $\mathcal{I}$) requires estimating the stable conditionals $P_{X_r|\mathbf{X}_{\text{pa}(r)}}$ (a.k.a. causal Markov kernels) in order to compute the interventional expectation via (11). For convenience (see §6 for details), here we opt for latent-variable implicit density models, but other conditional density estimation approaches may be also be used [e.g., 6, 8, 53]. Specifically, we model each conditional $p(x_r|\mathbf{x}_{\text{pa}(r)})$ with a conditional variational autoencoder (CVAE) [48] as:

$$p(x_r|\mathbf{x}_{\text{pa}(r)}) \approx p_{\psi_r}(x_r|\mathbf{x}_{\text{pa}(r)}) = \int p_{\psi_r}(x_r|\mathbf{x}_{\text{pa}(r)}, \mathbf{z}_r) p(\mathbf{z}_r) d\mathbf{z}_r, \qquad p(\mathbf{z}_r) := \mathcal{N}(\mathbf{0}, \mathbf{I}). \quad (12)$$

To facilitate sampling $x_r$ (and in analogy to the deterministic mechanisms $f_r$ in SCMs), we opt for deterministic decoders in the form of neural nets $D_r$ parametrised by $\psi_r$, i.e., $p_{\psi_r}(x_r|\mathbf{x}_{\text{pa}(r)}, \mathbf{z}_r) := \delta\big(x_r - D_r(\mathbf{x}_{\text{pa}(r)}, \mathbf{z}_r; \psi_r)\big)$, and rely on variational inference [60], amortised with approximate posteriors $q_{\phi_r}(\mathbf{z}_r|x_r, \mathbf{x}_{\text{pa}(r)})$ parametrised by encoders in the form of neural nets with parameters $\phi_r$. We learn both the encoder and decoder parameters by maximising the evidence lower bound (ELBO) using stochastic gradient descend [9, 26, 27, 40]. For further details, we refer to Appendix D.

**Remark 8.** *The collection of* CVAE*s can be interpreted as learning an approximate* SCM *of the form*

$$\mathcal{M}_{\text{CVAE}}: \qquad \mathbf{S} = \{X_r := D_r(\mathbf{X}_{pa(r)}, \mathbf{z}_r; \psi_r)\}_{r=1}^{d}, \qquad \mathbf{z}_r \sim \mathcal{N}(\mathbf{0}, \mathbf{I}) \quad \forall r \in [d] \quad (13)$$

*However, this family of* SCM*s may not allow to identify the true* SCM *(provided it can be expressed as above) from data without additional assumptions. Moreover, exact posterior inference over $\mathbf{z}_r$ given $\mathbf{x}^{F}$ is intractable, and we need to resort to approximations instead. It is thus unclear whether sampling from $q_{\phi_r}(\mathbf{z}_r|x_r^{F}, \mathbf{x}^{F}_{pa(r)})$ instead of from $p(\mathbf{z}_r)$ in (12) can be interpreted as a counterfactual within (13). For further discussion on such "pseudo-counterfactuals" we refer to Appendix C.*

# 6 Solving the probabilistic-recourse optimisation problems

We now discuss how to solve the resulting optimisation problems in (8) and (10). First, note that both problems differ only on the distribution over which the expectation in the constraint is taken: in (8) this is the counterfactual distribution of the descendants given in Proposition 6; and in (10) it is the interventional distribution identified in Proposition 7. In either case, computing the expectation for an arbitrary classifier $h$ is intractable. Here, we approximate these integrals via Monte Carlo by sampling $\mathbf{x}_{\mathrm{d}(\mathcal{I})}^{(m)}$ from the interventional or counterfactual distributions resulting from $a = do(\mathbf{X}_{\mathcal{I}} = \boldsymbol{\theta})$, i.e.,

$$\mathbb{E}_{\mathbf{X}_{\mathrm{d}(\mathcal{I})|\boldsymbol{\theta}}}\big[h\big(\mathbf{x}_{\mathrm{nd}(\mathcal{I})}^{\mathrm{F}}, \boldsymbol{\theta}, \mathbf{X}_{\mathrm{d}(\mathcal{I})}\big)\big] \approx \tfrac{1}{M} \sum_{m=1}^{M} h\big(\mathbf{x}_{\mathrm{nd}(\mathcal{I})}^{\mathrm{F}}, \boldsymbol{\theta}, \mathbf{x}_{\mathrm{d}(\mathcal{I})}^{(m)}\big).$$

**Brute-force approach.** A way to solve (8) and (10) is to (i) iterate over $a \in \mathbb{A}^{\mathrm{F}}$, with $\mathbb{A}^{\mathrm{F}}$ being a finite set of feasible actions (possibly as a result of discretising in the case of a continuous search space); (ii) approximately evaluate the constraint via Monte Carlo; and (iii) select a minimum cost action amongst all evaluated candidates satisfying the constraint. However, this may be computationally prohibitive and yield suboptimal interventions due to discretisation.

**Gradient-based approach.** Recall that, for actions of the form $a = do(\mathbf{X}_{\mathcal{I}} = \boldsymbol{\theta})$, we need to optimise over both the intervention *targets* $\mathcal{I}$ and the intervention *values* $\boldsymbol{\theta}$. Selecting targets is a hard combinatorial optimisation problem, as there are $2^{d'}$ possible choices for $d' \leq d$ actionable features, with a potentially infinite number of intervention values. We therefore consider different choices of targets $\mathcal{I}$ in parallel, and propose a gradient-based approach suitable for differentiable classifiers to efficiently find an optimal $\boldsymbol{\theta}$ for a given intervention set $\mathcal{I}$.[6] In particular, we first rewrite the constrained optimisation problem in unconstrained form with Lagrangian [23, 28]:

$$\mathcal{L}(\boldsymbol{\theta}, \lambda) := \mathrm{cost}^{\mathrm{F}}(a) + \lambda\big(\mathtt{thresh}(a) - \mathbb{E}_{\mathbf{X}_{\mathrm{d}(\mathcal{I})|\boldsymbol{\theta}}}\big[h\big(\mathbf{x}_{\mathrm{nd}(\mathcal{I})}^{\mathrm{F}}, \boldsymbol{\theta}, \mathbf{X}_{\mathrm{d}(\mathcal{I})}\big)\big]\big). \tag{14}$$

We then solve the saddle point problem $\min_{\boldsymbol{\theta}} \max_{\lambda} \mathcal{L}(\boldsymbol{\theta}, \lambda)$ arising from (14) with stochastic gradient descent [9, 26]. Since both the GP-SCM counterfactual (7) and the CVAE interventional distributions (12) admit a reparametrisation trick [27, 40], we can differentiate through the constraint:

$$\nabla_{\boldsymbol{\theta}} \mathbb{E}_{\mathbf{X}_{\mathrm{d}(\mathcal{I})}}\big[h\big(\mathbf{x}_{\mathrm{nd}(\mathcal{I})}^{\mathrm{F}}, \boldsymbol{\theta}, \mathbf{X}_{\mathrm{d}(\mathcal{I})}\big)\big] = \mathbb{E}_{\mathbf{z} \sim \mathcal{N}(\mathbf{0}, \mathbf{I})}\big[\nabla_{\boldsymbol{\theta}} h\big(\mathbf{x}_{\mathrm{nd}(\mathcal{I})}^{\mathrm{F}}, \boldsymbol{\theta}, \mathbf{x}_{\mathrm{d}(\mathcal{I})}(\mathbf{z})\big)\big]. \tag{15}$$

Here, $\mathbf{x}_{\mathrm{d}(\mathcal{I})}(\mathbf{z})$ is obtained by iteratively computing all descendants in topological order: either substituting $\mathbf{z}$ together with the other parents into the decoders $D_r$ for the CVAEs, or by using the Gaussian reparametrisation $x_r(\mathbf{z}) = \mu + \sigma \mathbf{z}$ with $\mu$ and $\sigma$ given by (7) for the GP-SCM. A similar gradient estimator for the variance which enters $\mathtt{thresh}(a)$ for $\gamma_{\mathrm{LCB}} \neq 0$ is derived in Appendix F.

# 7 Experimental results

In our experiments, we compare different approaches for *causal* algorithmic recourse on synthetic and semi-synthetic data sets. Additional results can be found in Apendix B.

**Compared methods.** We compare the naive point-based recourse approaches $\mathcal{M}_{\mathrm{LIN}}$ and $\mathcal{M}_{\mathrm{KR}}$ mentioned at the beginning of §4 as baselines with the proposed counterfactual GP-SCM $\mathcal{M}_{\mathrm{GP}}$ and the CVAE approach for sub-population-based recourse (CATE$_{\mathrm{CVAE}}$). For completeness, we also consider a CATE$_{\mathrm{GP}}$ approach as a GP can also be seen as modelling each conditional as a Gaussian,[7] and also evaluate the "pseudo-counterfactual" $\mathcal{M}_{\mathrm{CVAE}}$ approach discussed in Remark 8. Finally, we report oracle performance for individualised $\mathcal{M}_{\star}$ and sub-population-based recourse methods CATE$_{\star}$ by sampling counterfactuals and interventions from the true underlying SCM. We note that a comparison with non-causal recourse approaches that assume independent features [54, 44] or consider causal relations to generate counterfactual explanations but not recourse actions [19, 32] is neither natural nor straight-forward, because it is unclear whether descendant variables should be allowed to change, whether keeping their value constant should incur a cost, and, if so, how much, c.f. [22].

Table 1: Experimental results for the gradient-based approach on different 3-variable SCMs. We show average performance $\pm 1$ standard deviation for $N_{\text{runs}} = 100$, $N_{\text{MC-samples}} = 100$, and $\gamma_{\text{LCB}} = 2$.

| Method | LINEAR SCM | | | NON–LINEAR ANM | | | NON–ADDITIVE SCM | | |
|---|---|---|---|---|---|---|---|---|---|
| | Valid$_\star$ (%) | LCB | Cost (%) | Valid$_\star$ (%) | LCB | Cost (%) | Valid$_\star$ (%) | LCB | Cost (%) |
| $\mathcal{M}_\star$ | 100 | - | 10.9±7.9 | 100 | - | 20.1±12.3 | 100 | - | 13.2±11.0 |
| $\mathcal{M}_{\text{LIN}}$ | 100 | - | 11.0±7.0 | 54 | - | 20.6±11.0 | 98 | - | 14.0±13.5 |
| $\mathcal{M}_{\text{KR}}$ | 90 | - | 10.7±6.5 | 91 | - | 20.6±12.5 | 70 | - | 13.2±11.6 |
| $\mathcal{M}_{\text{GP}}$ | 100 | .55±.04 | 12.2±8.3 | 100 | .54±.03 | 21.9±12.9 | 95 | .52±.04 | 13.4±12.8 |
| $\mathcal{M}_{\text{CVAE}}$ | 100 | .55±.07 | 11.8±7.7 | 97 | .54±.05 | 22.6±12.3 | 95 | .51±.01 | 13.4±12.2 |
| CATE$_\star$ | 90 | .56±.07 | 11.9±9.2 | 97 | .55±.05 | 26.3±21.4 | 100 | .52±.02 | 13.5±13.0 |
| CATE$_{\text{GP}}$ | 93 | .56±.05 | 12.2±8.4 | 94 | .55±.06 | 25.0±14.8 | 94 | .52±.03 | 13.2±13.1 |
| CATE$_{\text{CVAE}}$ | 89 | .56±.08 | 12.1±8.9 | 98 | .54±.05 | 26.0±14.3 | 100 | .52±.05 | 13.6±12.9 |

**Metrics.** We compare recourse actions recommended by the different methods in terms of *cost*, computed as the L2-norm between the intervention $\boldsymbol{\theta}_{\mathcal{I}}$ and the factual value $\mathbf{x}_{\mathcal{I}}^{\text{F}}$, normalised by the range of each feature $r \in \mathcal{I}$ observed in the training data; and *validity*, computed as the percentage of individuals for which the recommended actions result in a favourable prediction under the true (oracle) SCM. For our probabilistic recourse methods, we also report the lower confidence bound LCB $:= \mathbb{E}[h] - \gamma_{\text{LCB}}\sqrt{\text{Var}[h]}$ of the selected action under the given method.

**Synthetic 3-variable SCMs under different assumptions.** In our first set of experiments, we consider three classes of SCMs over three variables with the same causal graph as in Fig. 1c. To test robustness of the different methods to assumptions about the form of the true structural equations, we consider a linear SCM, a non-linear ANM, and a more general, multi-modal SCM with non-additive noise. For further details on the exact form we refer to Appendix E.

Results are shown in Table 1. We observe that the point-based recourse approaches perform (relatively) well in terms of both validity and cost, when their underlying assumptions are met (i.e., $\mathcal{M}_{\text{LIN}}$ on the linear SCM and $\mathcal{M}_{\text{KR}}$ on the nonlinear ANM). Otherwise, validity significantly drops as expected (see, e.g., the results of $\mathcal{M}_{\text{LIN}}$ on the non-linear ANM, or of $\mathcal{M}_{\text{KR}}$ on the non-additive SCM). Moreover, we note that the inferior performance of $\mathcal{M}_{\text{KR}}$ compared to $\mathcal{M}_{\text{LIN}}$ on the linear SCM suggests an overfitting problem, which does not occur for its more conservative probabilistic counterpart $\mathcal{M}_{\text{GP}}$. Generally, the individualised approaches $\mathcal{M}_{\text{GP}}$ and $\mathcal{M}_{\text{CVAE}}$ perform very competitively in terms of cost and validity, especially on the linear and nonlinear ANMs. The subpopulation-based CATE approaches on the other hand, perform particularly well on the challenging non-additive SCM (on which the assumptions of GP approaches are violated) where CATE$_{\text{CVAE}}$ achieves perfect validity as the only non-oracle method. As expected, the subpopulation-based approaches generally lead to higher cost than the individualised ones, since the latter only aim to achieve recourse only for a given individual while the former do it for an entire group (see Fig. 2a).

**Semi-synthetic 7-variable SCM for loan-approval.** We also test our methods on a larger semi-synthetic SCM inspired by the German Credit UCI dataset [34]. We consider the variables age $A$, gender $G$, education-level $E$, loan amount $L$, duration $D$, income $I$, and savings $S$ with causal graph shown in Fig. 2b. We model age $A$, gender $G$ and loan duration $D$ as non-actionable variables, but consider $D$ to be mutable, i.e., it cannot be manipulated directly but is allowed to change (e.g., as a consequence of an intervention on $L$). The SCM includes linear and non-linear relationships, as well as different types of variables and noise distributions, and is described in more detail in Appendix E.

The results are summarised in Table 2, where we observe that the insights discussed above similarly apply for data generated from a more complex SCM, and for different classifiers. Finally, we show the influence of $\gamma_{\text{LCB}}$ on the performance of the proposed probabilistic approaches in Fig. 2c. We observe that lower values of $\gamma_{\text{LCB}}$ lead to lower validity (and cost), especially for the CATE approaches. As $\gamma_{\text{LCB}}$ increases validity approaches the corresponding oracles $\mathcal{M}_\star$ and CATE$_\star$, outperforming the point-based recourse approaches. In summary, our probabilistic recourse approaches are not only more robust, but also allow controlling the trade-off between validity and cost using $\gamma_{\text{LCB}}$.

Table 2: Experimental results for the 7-variable SCM for loan-approval. We show average performance $\pm 1$ standard deviation for $N_{\text{runs}} = 100$, $N_{\text{MC-samples}} = 100$, and $\gamma_{\text{LCB}} = 2.5$. For linear and non-linear logistic regression as classifiers, we use the gradient-based approach, whereas for the non-differentiable random forest classifier we rely on the brute-force approach (with 10 discretised bins per dimension) to solve the recourse optimisation problems.

| Method | LINEAR LOG. REGR. | | | NON-LIN. LOG. REGR. (MLP) | | | RANDOM FOREST(BRUTE-FORCE) | | |
|---|---|---|---|---|---|---|---|---|---|
| | Valid$_\star$ (%) | LCB | Cost (%) | Valid$_\star$ (%) | LCB | Cost (%) | Valid$_\star$ (%) | LCB | Cost (%) |
| $\mathcal{M}_\star$ | 100 | - | 15.8$\pm$7.6 | 100 | - | 11.0$\pm$7.0 | 100 | - | 15.2$\pm$7.5 |
| $\mathcal{M}_{\text{LIN}}$ | 19 | - | 15.4$\pm$7.4 | 80 | - | 11.0$\pm$6.9 | 94 | - | 15.6$\pm$7.6 |
| $\mathcal{M}_{\text{KR}}$ | 41 | - | 15.6$\pm$7.5 | 87 | - | 11.1$\pm$7.0 | 92 | - | 15.1$\pm$7.4 |
| $\mathcal{M}_{\text{GP}}$ | 100 | .50$\pm$.00 | 18.0$\pm$7.7 | 100 | .52$\pm$.04 | 11.7$\pm$7.3 | 100 | .66$\pm$.14 | 16.3$\pm$7.4 |
| $\mathcal{M}_{\text{CVAE}}$ | 100 | .50$\pm$.00 | 16.6$\pm$7.6 | 99 | .51$\pm$.01 | 11.3$\pm$6.9 | 100 | .66$\pm$.14 | 15.9$\pm$7.4 |
| CATE$_\star$ | 93 | .50$\pm$.01 | 22.0$\pm$9.4 | 95 | .52$\pm$.05 | 12.0$\pm$7.7 | 98 | .66$\pm$.15 | 17.0$\pm$7.3 |
| CATE$_{\text{GP}}$ | 93 | .50$\pm$.02 | 21.7$\pm$9.2 | 93 | .51$\pm$.06 | 12.0$\pm$7.4 | 100 | .67$\pm$.15 | 17.1$\pm$7.4 |
| CATE$_{\text{CVAE}}$ | 94 | .49$\pm$.01 | 23.7$\pm$11.3 | 95 | .51$\pm$.03 | 12.0$\pm$7.8 | 100 | .68$\pm$.15 | 17.9$\pm$7.4 |

## 8 Discussion

**Assumptions, limitations, and extensions.** Throughout the paper, we have assumed a known causal graph and causal sufficiency. While this may not hold for all settings, it is the minimal necessary set of assumptions for causal reasoning from observational data alone. Access to instrumental variables or experimental data may help further relax these assumptions [3, 11, 50]. Moreover, if only a partial graph is available or some relations are known to be confounded, one will need to restrict recourse actions to the subset of interventions that are still identifiable [45, 46, 51]. An alternative approach could address causal sufficiency violations by relying on latent variable models to estimate confounders from multiple causes [61] or proxy variables [31], or to work with bounds on causal effects instead [4, 49]. We relegate the investigation of these settings to future work.

**On the counterfactual vs interventional nature of recourse.** Given that we address two different notions of recourse—counterfactual/individualised (rung 3) vs. interventional/subpopulation-based (rung 2)—one may ask which framing is more appropriate. Since the main difference is whether the background variables $\mathbf{U}$ are assumed fixed (counterfactual) or not (interventional) when reasoning about actions, we believe that this question is best addressed by thinking about the type of environment and interpretation of $\mathbf{U}$: if the environment is static, or if $\mathbf{U}$ (mostly) captures unobserved information about the individual, the counterfactual notion seems to be the right one; if, on the other hand, $\mathbf{U}$ also captures environmental factors which may change, e.g., between consecutive loan applications, then the interventional notion of recourse may be more appropriate. In practice, both notions may be present (for different variables), and the proposed approaches can be combined depending on the available domain knowledge since each parent-child causal relation is treated separately. We emphasise that the subpopulation-based approach is also practically motivated by a reluctance to make (parametric) assumptions about the structural equations which are untestable but necessary for counterfactual reasoning. It may therefore be useful to avoid problems of misspecification, even for counterfactual recourse, as demonstrated experimentally for the non-additive SCM.

## 9 Conclusion

In this work, we studied the problem of algorithmic recourse from a causal perspective. As negative result, we first showed that algorithmic recourse cannot be guaranteed in the absence of perfect knowledge about the underlying SCM governing the world, which unfortunately is not available in practice. To address this limitation, we proposed two probabilistic approaches to achieve recourse under more realistic assumptions. In particular, we derived i) an individual-level recourse approach based on GPs that approximates the counterfactual distribution by averaging over the family of additive Gaussian SCMs; and ii) a subpopulation-based approach, which assumes that only the causal graph is known and makes use of CVAEs to estimate the conditional average treatment effect of an intervention on a subpopulation similar to the individual seeking recourse. Our experiments showed that the proposed probabilistic approaches not only result in more robust recourse interventions than approaches based on point estimates of the SCM, but also allows to trade-off validity and cost.

## Broader Impact

Our work falls into the domain of explainable AI, which—given the increasing use of often intransparent ("blackbox") machine learning models in consequential decision making—is of rapidly-growing societal importance. In particular, we consider the task of enabling and facilitating *algorithmic recourse*, which aims to provide individuals with guidance and recommendations on how best (i.e., efficiently and ideally at low cost) to recover from unfavourable decisions made by an automated system. To address this task, we build on the framework of causal modelling, which constitutes a principled and mathematically rigorous way to reason about the downstream effects of actions. Since correlation does not imply causation, this requires to make additional assumptions based on a general understanding of the domain at hand. While this may perhaps seem restrictive at first, we point out that other approaches to explainability also make implicit assumptions of a causal nature (e.g., that all features can be changed at will without affecting others in the case of "counterfactual" explanations), without explicitly and clearly stating such assumptions. The advantage of phrasing assumptions about relations between features in the form of a causal graph is that the latter is transparent and intuitive to understand and can thus be challenged by decision makers and individuals alike.

While theoretically sound from a causal perspective, at the same time, our method is aimed at being practical by not making further assumptions beyond the causal graph which would be hard or impossible to test or challenge empirically—in contrast to the assumed known specification of the full SCM in [22]. We start from the position that the model is only partially known, and use this to motivate probabilistic approaches to causal algorithmic recourse which take uncertainty into account. Our approaches are more robust to misspefication than naive point-based recourse methods (as demonstrated experimentally): "system-failure" is thus fundamentally baked in to our methods. Moreover, the interpretable "conservativeness parameter" $\gamma_{\text{LCB}}$ can be used trade-off the desired level of robustness against the effort an individual is willing to put into achieving recourse.

The importance of causal reasoning for an ethical and socially beneficial use of ML-assisted technology has also been stressed in a number of recent works in the field of explainability and fair algorithmic decision making [29, 42, 24, 63, 64, 10, 57, 15]. We thus hope that some of the probabilistic approaches for causal reasoning under imperfect knowledge proposed in this work may also prove useful for related tasks such as fairness, accountability, transparency. To this end, we have created a user-friendly implementation of all the approaches proposed in this work that we will make publicly available to be scrutinised, re-used, and further improved by the community. The code is highly flexible and only requires the specification of a causal graph, as well as a labelled training dataset.

Since our work considers the classifier as given, it is possible that it is explicitly discriminatory or reproduces biases in the data. While not directly addressing this problem, our work aims to enable individuals to overcome a potentially unfairly obtained decision with minimal effort. If successful recourse examples are included in future training data, this may help de-bias a system over time; we consider the intersection of our work with fair decision making in the context of a classifier evolving over time as the result of further data collection [25] a fruitful and important direction for future research. In addition, observing that certain minority groups consistently receive more costly recourse recommendations may be a way to reveal bias in the underlying decision making system.

While our framework is intended to help individuals increase their chances for a more favourable prediction given that they were, e.g., denied a loan or bail, we cannot rule out a priori, that the same approach could also be used by foes in unintended ways, e.g., to "game" a spam filter or similar system built to protect society from harm. However, since our framework requires the specification of a causal graph which usually requires an understanding of the domain and the causal influences at play, it is unlikely that it could be abused by a purely virtual system without a human in the loop.

## Acknowledgments and Disclosure of Funding

The authors would like to thank Adrian Weller, Floyd Kretschmar, Junhyung Park, Matthias Bauer, Miriam Rateike, Nicolo Ruggeri, Umang Bhatt, and Vidhi Lalchand for helpful feedback and discussions. Moreover, a special thanks to Adrià Garriga-Alonso for insightful input on some of the GP-derivations and to Adrián Javaloy Bornás for invaluable help with the CVAE-training. AHK acknowledges NSERC and CLS for generous funding support.

## Footnotes

[2]Also known as non-parametric structural equation model with independent errors (NPSEM-IE).

[3]I.e., for $r \in [d]$, $P_{X_r | \mathbf{X}_{\mathrm{pa}(r)}}(X_r | \mathbf{X}_{\mathrm{pa}(r)}) := P_{U_r}(f_r^{-1}(X_r | \mathbf{X}_{\mathrm{pa}(r)}))$, where $f_r^{-1}(X_r | \mathbf{X}_{\mathrm{pa}(r)})$ denotes the pre-image of $X_r$ given $\mathbf{X}_{\mathrm{pa}(r)}$ under $f_r$, i.e., $f_r^{-1}(X_r | \mathbf{X}_{\mathrm{pa}(r)}) := \{u \in \mathcal{U}_r : f_r(\mathbf{X}_{\mathrm{pa}(r)}, u) = X_r\}$.

[4]Following the related literature, we consider a binary classification task by convention; most of our considerations extend to multi-class classification or regression settings as well though.

[5]This follows from abduction on $\mathbf{x}^{\mathrm{F}} = (1, 0, 0)$ which for both $\mathcal{M}_A$ and $\mathcal{M}_B$ implies $U_3 = 0$.

[6]For large $d$ when enumerating all $\mathcal{I}$ becomes computationally prohibitive, we can upper-bound the allowed number of variables to be intervened on simultaneously (e.g., $|\mathcal{I}| \leq 3$), or choose a greedy approach to select $\mathcal{I}$.

[7]Sampling from the noise prior instead of the posterior in (6) leads to an interventional distribution in (7).

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
