[Supplementary Material]

# A  Proofs

## A.1  Proof of Proposition 5

**Proposition 5** (GP-SCM noise posterior). *Let $\{\mathbf{x}^i\}_{i=1}^n$ be an observational sample from (5). For each $r \in [d]$ with non empty parent set $|pa(r)| > 0$, the posterior distribution of the noise vector $\mathbf{u}_r = (u_r^1, ..., u_r^n)$, conditioned on $\mathbf{x}_r = (x_r^1, ..., x_r^n)$ and $\mathbf{X}_{pa(r)} = (\mathbf{x}_{pa(r)}^1, ..., \mathbf{x}_{pa(r)}^n)$, is given by*

$$\mathbf{u}_r | \mathbf{X}_{pa(r)}, \mathbf{x}_r \sim \mathcal{N}\left(\sigma_r^2 (\mathbf{K} + \sigma_r^2 \mathbf{I})^{-1} \mathbf{x}_r, \sigma_r^2 \left(\mathbf{I} - \sigma_r^2 (\mathbf{K} + \sigma_r^2 \mathbf{I})^{-1}\right)\right), \tag{6}$$

*where $\mathbf{K} := \left(k_r\left(\mathbf{x}_{pa(r)}^i, \mathbf{x}_{pa(r)}^j\right)\right)_{ij}$ denotes the Gram matrix.*

*Proof.* First, note that, by definition, $\mathbf{u}_r$ is independent of $\mathbf{f}_r = (f_r(\mathbf{x}_{pa(r)}^1), ..., f_r(\mathbf{x}_{pa(r)}^n))$ given $\mathbf{X}_{pa(r)}$. Moreover, it follows from the assumed GP-SCM model in (5) and Definition 4, as well as properties of the GP prior, that both are multivariate Gaussian random variables with distributions given by

$$\mathbf{u}_r \sim \mathcal{N}(\mathbf{0}, \sigma_r^2 \mathbf{I}) \quad \text{independently of} \quad \mathbf{X}_{pa(r)}, \quad \text{and} \tag{A.1}$$

$$\mathbf{f}_r | \mathbf{X}_{pa(r)} \sim \mathcal{N}(\mathbf{0}, \mathbf{K}), \tag{A.2}$$

where $\mathbf{0}$ denotes the zero vector (or matrix, see below) and $\mathbf{K}$ is as defined in Proposition 5.

Since independent multivariate Gaussian random variables are jointly multivariate Gaussian, we thus have

$$\begin{pmatrix} \mathbf{u}_r \\ \mathbf{f}_r \end{pmatrix} | \mathbf{X}_{pa(r)} \sim \mathcal{N}(\mathbf{0}, \Sigma), \quad \text{where} \quad \Sigma = \begin{pmatrix} \sigma_r^2 \mathbf{I} & \mathbf{0} \\ \mathbf{0} & \mathbf{K} \end{pmatrix} \tag{A.3}$$

Noting that $\mathbf{x}_r = \mathbf{f}_r + \mathbf{u}_r$ and applying a linear transformation to (A.3), we then obtain

$$\begin{pmatrix} \mathbf{u}_r \\ \mathbf{x}_r \end{pmatrix} | \mathbf{X}_{pa(r)} = \begin{pmatrix} \mathbf{I} & \mathbf{0} \\ \mathbf{I} & \mathbf{I} \end{pmatrix} \begin{pmatrix} \mathbf{u}_r \\ \mathbf{f}_r \end{pmatrix} | \mathbf{X}_{pa(r)} \sim \mathcal{N}(\mathbf{0}, \tilde{\Sigma}), \quad \text{where} \quad \tilde{\Sigma} = \begin{pmatrix} \sigma_r^2 \mathbf{I} & \sigma_r^2 \mathbf{I} \\ \sigma_r^2 \mathbf{I} & \mathbf{K} + \sigma_r^2 \mathbf{I} \end{pmatrix}. \tag{A.4}$$

Conditioning on $\mathbf{x}_r$ and using the conditioning formula [e.g., 52], the result follows:

$$\mathbf{u}_r | \mathbf{X}_{pa(r)}, \mathbf{x}_r \sim \mathcal{N}\left(\mathbf{0} + \sigma_r^2 \mathbf{I}(\mathbf{K} + \sigma_r^2 \mathbf{I})^{-1}(\mathbf{x}_r - \mathbf{0}), \sigma_r^2 \mathbf{I} - \sigma_r^2 \mathbf{I}(\mathbf{K} + \sigma_r^2 \mathbf{I})^{-1} \sigma_r^2 \mathbf{I}\right) \tag{A.5}$$

$$\sim \mathcal{N}\left(\sigma_r^2 (\mathbf{K} + \sigma_r^2 \mathbf{I})^{-1} \mathbf{x}_r, \sigma_r^2 \left(\mathbf{I} - \sigma_r^2 (\mathbf{K} + \sigma_r^2 \mathbf{I})^{-1}\right)\right) \tag{A.6}$$

$\square$

## A.2  Proof of Proposition 6

**Proposition 6** (GP-SCM counterfactual distribution). *Let $\{\mathbf{x}^i\}_{i=1}^n$ be an observational sample from (5). Then, for $r \in [d]$ with $|pa(r)| > 0$, the counterfactual distribution over $X_r$ had $\mathbf{X}_{pa(r)}$ been $\tilde{\mathbf{x}}_{pa(r)}$ (instead of $\mathbf{x}_{pa(r)}^F$) for individual $\mathbf{x}^F \in \{\mathbf{x}^i\}_{i=1}^n$ is given by*

$$X_r(\mathbf{X}_{pa(r)} = \tilde{\mathbf{x}}_{pa(r)}) | \mathbf{x}^F, \{\mathbf{x}^i\}_{i=1}^n \sim \mathcal{N}\left(\mu_r^F + \tilde{\mathbf{k}}^T (\mathbf{K} + \sigma_r^2 \mathbf{I})^{-1} \mathbf{x}_r, s_r^F + \tilde{k} - \tilde{\mathbf{k}}^T (\mathbf{K} + \sigma_r^2 \mathbf{I})^{-1} \tilde{\mathbf{k}}\right), \tag{7}$$

*where $\tilde{k} := k_r(\tilde{\mathbf{x}}_{pa(r)}, \tilde{\mathbf{x}}_{pa(r)})$, $\tilde{\mathbf{k}} := \left(k_r(\tilde{\mathbf{x}}_{pa(r)}, \mathbf{x}_{pa(r)}^1), ..., k_r(\tilde{\mathbf{x}}_{pa(r)}, \mathbf{x}_{pa(r)}^n)\right)$, $\mathbf{x}_r$ and $\mathbf{K}$ as defined in Proposition 5, and $\mu_r^F$ and $s_r^F$ are the posterior mean and variance of $u_r^F$ given by (6).*

*Proof.* We follow the three steps of abduction, action, and prediction for computing counterfactual distributions (see §2 for more details). Starting from the factual observation $\mathbf{x}^F \in \{x^i\}_{i=1}^n$ generated according to

$$x_r^F := f_r(\mathbf{x}_{pa(r)}^F) + u_r^F, \tag{A.7}$$

we first compute the noise posterior (*abduction*). According to Proposition 5 it is given by a marginal of (6), i.e.,

$$u_r^F | \mathbf{X}_{pa(r)}, \mathbf{x}_r \sim \mathcal{N}(\mu_r^F, s_r^F) \tag{A.8}$$

where $\mu_r^F$ is given by element F of the mean vector

$$\boldsymbol{\mu}_r = \sigma_r^2 (\mathbf{K} + \sigma_r^2 \mathbf{I})^{-1} \mathbf{x}_r \tag{A.9}$$

and $s_r^F$ is given by element $(\mathrm{F}, \mathrm{F})$ of the covariance matrix

$$S_r = \sigma_r^2 \left(\mathbf{I} - \sigma_r^2 (\mathbf{K} + \sigma_r^2 \mathbf{I})^{-1}\right) \tag{A.10}$$

of the noise posterior given by (6).

Next, we simulate the hypothetical intervention by updating the structural equation (A.7) (*action step*),

$$x_r^F(\mathbf{X}_{pa(r)} = \tilde{\mathbf{x}}_{pa(r)}) := f_r(\tilde{x}_{pa(r)}) + u_r^F. \tag{A.11}$$

The GP predictive posterior at the new input $\tilde{x}_{pa(r)}$ has distribution [see, e.g., 62],

$$f_r(\tilde{x}_{pa(r)}) | \mathbf{X}_{pa(r)}, \mathbf{x}_r \sim \mathcal{N}(\tilde{\mathbf{k}}^T (\mathbf{K} + \sigma_r^2 \mathbf{I})^{-1} \mathbf{x}_r, \tilde{k} - \tilde{\mathbf{k}}^T (\mathbf{K} + \sigma_r^2 \mathbf{I})^{-1} \tilde{\mathbf{k}}). \tag{A.12}$$

Substituting (A.12) and (A.8) into (A.11) and noting that the sum of two Gaussians is again Gaussian with mean and variance equal to the sums of means and variances of the two individual Gaussians (*prediction step*) completes the proof. $\square$

## A.3 Proof of Proposition 7

**Proposition 7.** *Subject to causal sufficiency, $P_{\mathbf{X}_{d(\mathcal{I})}|do(\mathbf{X}_{\mathcal{I}}=\boldsymbol{\theta}),\mathbf{x}_{nd(\mathcal{I})}^{F}}$ is observationally identifiable:*

$$p\big(\mathbf{X}_{d(\mathcal{I})}|do(\mathbf{X}_{\mathcal{I}}=\boldsymbol{\theta}),\mathbf{x}_{nd(\mathcal{I})}^{F}\big) = \prod_{r\in d(\mathcal{I})} p\left(X_r|\mathbf{X}_{pa(r)}\right)\Big|_{\mathbf{X}_{\mathcal{I}}=\boldsymbol{\theta},\mathbf{X}_{nd(\mathcal{I})}=\mathbf{x}_{nd(\mathcal{I})}^{F}}. \tag{11}$$

*Proof.* This is a direct consequence of the properties of causally sufficient (Markovian) causal models, but we include a derivation for completeness. Recall that $P$ factorises over its underlying causal graph $\mathcal{G}$ as follows,

$$p(\mathbf{X}) = \prod_{r\in[d]} p(X_r|\mathbf{X}_{\mathrm{pa}(r)}). \tag{A.13}$$

This joint distribution is transformed by the intervention $do(\mathbf{X}_{\mathcal{I}}=\boldsymbol{\theta})$ as follows,

$$P(\mathbf{X}_{-\mathcal{I}}, do(\mathbf{X}_{\mathcal{I}}=\boldsymbol{\theta})) = \delta(\mathbf{X}_{\mathcal{I}}=\boldsymbol{\theta}) \prod_{r\in[d]\setminus\mathcal{I}} P(X_r|\mathbf{X}_{\mathrm{pa}(r)}). \tag{A.14}$$

Splitting the non-intervened variables into descendants $\mathrm{d}(\mathcal{I})$ and non-descendants $\mathrm{nd}(\mathcal{I})$, and conditioning on the intervened variables $do(\mathbf{X}_{\mathcal{I}}=\boldsymbol{\theta})$, we obtain

$$P(\mathbf{X}_{\mathrm{nd}(\mathcal{I})}, \mathbf{X}_{\mathrm{d}(\mathcal{I})}|do(\mathbf{X}_{\mathcal{I}}=\boldsymbol{\theta})) = \left(\prod_{r\in\mathrm{nd}(\mathcal{I})\cup\mathrm{d}(\mathcal{I})} P(X_r|\mathbf{X}_{\mathrm{pa}(r)})\right)\Bigg|_{\mathbf{X}_{\mathcal{I}}=\boldsymbol{\theta}}. \tag{A.15}$$

As the non-descendants $\mathbf{X}_{\mathrm{nd}(\mathcal{I})}$ are, by their very definition, not affected by the intervention, we can write

$$P(\mathbf{X}_{\mathrm{nd}(\mathcal{I})}, \mathbf{X}_{\mathrm{d}(\mathcal{I})}|do(\mathbf{X}_{\mathcal{I}}=\boldsymbol{\theta})) = \left(\prod_{r\in\mathrm{d}(\mathcal{I})} P(X_r|\mathbf{X}_{\mathrm{pa}(r)})\right)\Bigg|_{\mathbf{X}_{\mathcal{I}}=\boldsymbol{\theta}} \prod_{r\in\mathrm{nd}(\mathcal{I})} P(X_r|\mathbf{X}_{\mathrm{pa}(r)}).$$

We can thus condition on a particular value of $\mathbf{X}_{\mathrm{nd}(\mathcal{I})}$ to obtain

$$P\left(\mathbf{X}_{\mathrm{d}(\mathcal{I})}|do(\mathbf{X}_{\mathcal{I}}=\boldsymbol{\theta}), \mathbf{X}_{\mathrm{nd}(\mathcal{I})}=\mathbf{x}_{\mathrm{nd}(\mathcal{I})}^{\mathrm{F}}\right) = \left(\prod_{r\in\mathrm{d}(\mathcal{I})} P(X_r|\mathbf{X}_{pa(r)})\right)\Bigg|_{\mathbf{X}_{\mathcal{I}}=\boldsymbol{\theta},\mathbf{X}_{\mathrm{nd}(\mathcal{I})}=\mathbf{x}_{\mathrm{nd}(\mathcal{I})}^{\mathrm{F}}} \tag{A.16}$$

$\square$

# B  Additional results

This section presents additional results complementing those from Section 7. Table 3 presents results that mirror those in Table 1, where the brute-force approach discussed at the beginning of §6 is used instead of the gradient-based optimisation. Here, each real-valued feature was discretised into 20 bins within the range of its observed values in the training dataset.

Fig. 3 mirrors the results in Fig. 2c, for which a snapshot ($\gamma_{\mathrm{LCB}} = 2.5$) is also provided in Table 2. Here we show the trade-off between validity and cost by varying the values of $\gamma_{\mathrm{LCB}}$, using as trained classifiers a non-linear multilayer perceptron (MLP) in (a) and a non-differentiable random forest classifer in (b). Note that optimisation for the latter can only be done with the brute-force approach. All these additional results mostly confirm the insights presented in the main body.

Finally, Table 4 provides a qualitative comparison of the proposed recourse approaches against the oracles and baselines in terms of their selection of intervention targets. We show empirically, on the three synthetic datasets, that CATE approaches have more predictable behaviour, as they are less sensitive to model assumptions, and are thus more preferable for the individual seeking recourse under imperfect causal knowledge.

Table 3: Experimental results for the brute-force (20-bin discretization) approach on different 3-variable SCMs. We show average performance for $N_{\mathrm{runs}} = 100$, $N_{\mathrm{MC\text{-}samples}} = 100$, and $\gamma_{\mathrm{LCB}} = 2$. The relative trends reflect those in Table 1.

| Method | LINEAR SCM | | | NON-LINEAR ANM | | | NON-ADDITIVE SCM | | |
|---|---|---|---|---|---|---|---|---|---|
| | Valid$_\star$ (%) | LCB | Cost (%) | Valid$_\star$ (%) | LCB | Cost (%) | Valid$_\star$ (%) | LCB | Cost (%) |
| $\mathcal{M}_\star$ | 100 | - | 11.0±5.6 | 100 | - | 20.7±11.0 | 100 | - | 15.8± 8.9 |
| $\mathcal{M}_{\mathrm{LIN}}$ | 100 | - | 11.3±5.8 | 60 | - | 19.9± 8.9 | 92 | - | 17.0±10.4 |
| $\mathcal{M}_{\mathrm{KR}}$ | 95 | - | 11.2±5.6 | 88 | - | 20.5±10.7 | 47 | - | 15.8±10.6 |
| $\mathcal{M}_{\mathrm{GP}}$ | 100 | .55±.04 | 11.6±5.8 | 99 | .55±.04 | 21.2±10.9 | 88 | .58±.05 | 16.8±10.3 |
| $\mathcal{M}_{\mathrm{CVAE}}$ | 100 | .55±.04 | 11.5±5.8 | 95 | .55±.03 | 21.7±10.7 | 95 | .59±.07 | 16.9±10.3 |
| CATE$_\star$ | 90 | .57±.07 | 11.0±5.5 | 95 | .55±.05 | 22.8±10.8 | 99 | .57±.06 | 16.2± 8.9 |
| CATE$_{\mathrm{GP}}$ | 92 | .56±.07 | 11.2±5.5 | 95 | .55±.04 | 22.8±10.9 | 85 | .58±.07 | 16.4±10.5 |
| CATE$_{\mathrm{CVAE}}$ | 90 | .57±.06 | 11.1±5.4 | 96 | .55±.03 | 23.0±10.8 | 94 | .59±.07 | 16.8±10.2 |

(a) MLP                                    (b) random forest

Figure 3: Trade-off between validity and cost which can be controlled via $\gamma_{\mathrm{LCB}}$ for the probabilistic recourse methods. Shown is the same setting as in Fig. 2c using instead a non-linear logistic regression in the form of a multilayer perceptron (MLP; left), and a random forest (right) as classifiers $h$.

Table 4: Experimental results for the gradient-descent approach on different 3-variable SCMs (top to bottom: linear SCM, non-linear ANM, non-additive SCM). We show average performance for $N_{\text{runs}} = 100$, $N_{\text{MC-samples}} = 100$, and $\gamma_{\text{LCB}} = 2$, and display the number (out of $N_{\text{runs}}$) of performed interventions on all subsets of variables by each recourse type. The two right-most columns display how many of the intervention sets for each recourse type agreed with the suggestions made by the oracle methods, $\mathcal{M}_\star$ and CATE$_\star$, respectively. We observe that interventions proposed by the subpopulation-based oracle often differ from the ones proposed at the individual level, which can be visually explained by Fig. 2a. Importantly, we observe general agreement among all CATE approaches in their selection of intervened-upon variables. In contrast, we observe that individual-based methods deviate away from their oracle (i.e., $\mathcal{M}_\star$) in their selection of variables to intervene upon for recourse. This result further suggest that the CATE approaches presented in this work exhibit more predictable behaviour, as they are less sensitive to model assumptions, and are thus more preferable for the individual seeking recourse under imperfect causal knowledge.

| Method | SCM | | | INTERVENTION SET | | | | | | | IDENTICAL INT. SET | |
|---|---|---|---|---|---|---|---|---|---|---|---|---|
| | Valid$_\star$ (%) | LCB | Cost (%) | $\{X_1\}$ | $\{X_2\}$ | $\{X_3\}$ | $\{X_1,X_2\}$ | $\{X_1,X_3\}$ | $\{X_2,X_3\}$ | $\{X_1,X_2,X_3\}$ | $\mathcal{M}_\star$ | CATE$_\star$ |
| $\mathcal{M}_\star$ | 100 | - | 10.9±7.9 | 0 | 25 | 0 | 56 | 0 | 0 | 19 | 100 | 23 |
| $\mathcal{M}_{\text{LIN}}$ | 100 | - | 11.0±7.0 | 0 | 26 | 0 | 50 | 0 | 1 | 23 | 52 | 23 |
| $\mathcal{M}_{\text{KR}}$ | 90 | - | 10.7±6.5 | 0 | 22 | 0 | 44 | 0 | 0 | 34 | 54 | 27 |
| $\mathcal{M}_{\text{GP}}$ | 100 | .55±.04 | 12.2±8.3 | 0 | 6 | 0 | 13 | 0 | 7 | 74 | 25 | 61 |
| $\mathcal{M}_{\text{CVAE}}$ | 100 | .55±.07 | 11.8±7.7 | 0 | 12 | 0 | 25 | 0 | 5 | 58 | 31 | 57 |
| CATE$_\star$ | 90 | .56±.07 | 11.9±9.2 | 0 | 6 | 0 | 11 | 0 | 13 | 70 | 23 | 100 |
| CATE$_{\text{GP}}$ | 93 | .56±.05 | 12.2±8.4 | 0 | 3 | 0 | 9 | 1 | 15 | 72 | 18 | 76 |
| CATE$_{\text{CVAE}}$ | 89 | .56±.08 | 12.1±8.9 | 0 | 6 | 1 | 11 | 0 | 16 | 66 | 18 | 78 |
| $\mathcal{M}_\star$ | 100 | - | 20.1±12.3 | 70 | 0 | 0 | 2 | 16 | 0 | 11 | 99 | 17 |
| $\mathcal{M}_{\text{LIN}}$ | 54 | - | 20.6±11.0 | 13 | 0 | 0 | 0 | 81 | 0 | 5 | 20 | 41 |
| $\mathcal{M}_{\text{KR}}$ | 91 | - | 20.6±12.5 | 65 | 0 | 0 | 1 | 23 | 0 | 10 | 76 | 22 |
| $\mathcal{M}_{\text{GP}}$ | 100 | .54±.03 | 21.9±12.9 | 39 | 0 | 0 | 0 | 38 | 0 | 22 | 54 | 38 |
| $\mathcal{M}_{\text{CVAE}}$ | 97 | .54±.05 | 22.6±12.3 | 33 | 0 | 0 | 0 | 51 | 0 | 15 | 45 | 42 |
| CATE$_\star$ | 97 | .55±.05 | 26.3±21.4 | 4 | 0 | 0 | 0 | 44 | 2 | 49 | 17 | 99 |
| CATE$_{\text{GP}}$ | 94 | .55±.06 | 25.0±14.8 | 4 | 1 | 0 | 0 | 37 | 4 | 53 | 11 | 69 |
| CATE$_{\text{CVAE}}$ | 98 | .54±.05 | 26.0±14.3 | 3 | 0 | 0 | 1 | 32 | 1 | 62 | 12 | 70 |
| $\mathcal{M}_\star$ | 100 | - | 13.2±11.0 | 0 | 0 | 1 | 0 | 11 | 78 | 7 | 97 | 78 |
| $\mathcal{M}_{\text{LIN}}$ | 98 | - | 14.0±13.5 | 0 | 0 | 0 | 1 | 0 | 85 | 11 | 81 | 77 |
| $\mathcal{M}_{\text{KR}}$ | 70 | - | 13.2±11.6 | 0 | 17 | 0 | 4 | 10 | 59 | 7 | 55 | 53 |
| $\mathcal{M}_{\text{GP}}$ | 95 | .52±.04 | 13.4±12.8 | 3 | 1 | 2 | 0 | 0 | 82 | 9 | 73 | 78 |
| $\mathcal{M}_{\text{CVAE}}$ | 95 | .51±.01 | 13.4±12.2 | 0 | 3 | 1 | 5 | 2 | 71 | 15 | 72 | 76 |
| CATE$_\star$ | 100 | .52±.02 | 13.5±13.0 | 0 | 0 | 2 | 0 | 9 | 77 | 9 | 78 | 97 |
| CATE$_{\text{GP}}$ | 94 | .52±.03 | 13.2±13.1 | 3 | 1 | 5 | 0 | 3 | 73 | 12 | 70 | 76 |
| CATE$_{\text{CVAE}}$ | 100 | .52±.05 | 13.6±12.9 | 0 | 1 | 2 | 0 | 1 | 82 | 11 | 78 | 78 |

## C    (Non-)identifability of SCMs under different assumptions

In general form, i.e., without any further assumption on the structural equations $\mathbf{S}$ or noise distribution $P_{\mathbf{U}}$, SCMs are not identifiable from data alone, meaning that there are multiple different SCMs (possibly with different underlying causal graphs) which imply the same observational distribution [38]. One possible construction relies on the use of the inverse cumulative distribution function (cdf) in combination with uniformly-distributed random variables [12] and is also used in non-identifiability proofs for non-linear independent component analysis (ICA) [17]. Even knowing the causal graph is generally not enough as summarised in the following proposition.

**Proposition 9.** *Even when the causal graph is known, the conditionals $P(X_r|\mathbf{X}_{pa(r)})$ alone are insufficient to uniquely determine the structural equations $X_r := f_r(\mathbf{X}_{pa(r)}, U_r)$ without further assumptions.*

*Proof.* This can be shown by using the following argument from [18, Footnote 1] (adapted to our notation):

> "*let $U_r$ consist of (possibly uncountably many) real-valued random variables $U_r[\mathbf{x}_{pa(r)}]$, one for each value $\mathbf{x}_{pa(r)}$ of the parents $\mathbf{X}_{pa(r)}$. Let $U_r[\mathbf{x}_{pa(r)}]$ be distributed according to $P_{X_r|\mathbf{x}_{pa(r)}}$ and define $f_r(\mathbf{x}_{pa(r)}, U_r) := U_r[\mathbf{x}_{pa(r)}]$. Then $X_r|\mathbf{X}_{pa(r)}$ has distribution $P_{X_r|\mathbf{X}_{pa(r)}}$*".

We can now build on this formulation to construct a second SCM with the same observational distribution and causal graph, e.g., by shifting the noise variables and structural equations by some fixed constant $C$ as follows.

For $r \in [d]$, define $Y_r := X_r - C$. Let $\tilde{U}_r$ consist of (possibly uncountably many) real-valued random variables $\tilde{U}_r[\mathbf{x}_{pa(r)}]$, one for each value $\mathbf{x}_{pa(r)}$ of the parents $\mathbf{X}_{pa(r)}$. Let $\tilde{U}_r[\mathbf{x}_{pa(r)}]$ be distributed according to $P_{Y_r|\mathbf{x}_{pa(r)}}$ and define $f_r(\mathbf{x}_{pa(r)}, \tilde{U}_r) := \tilde{U}_r[\mathbf{x}_{pa(r)}] + C$. Then $X_r|\mathbf{X}_{pa(r)}$ also has distribution $P_{X_r|\mathbf{X}_{pa(r)}}$, but for $C \neq 0$ the structural equations and noise distributions are different from the previous construction. $\square$

In the case of the CVAE-SCM model from (13) the setting is slightly less general than the above, since we additionally assume that: (i) the noise distributions are isotropic multivariate Gaussian distributions of fixed dimension, $\mathbf{z}_r \sim \mathcal{N}_{d_{\mathbf{z}_r}}(\mathbf{0}, \mathbf{I})$; and (ii) the structural equations $D_r$ are from the class of functions that can be expressed as feedforward neural networks if fixed width and depth with learnable parameters $\psi_r$.

Unfortunately, we are not aware of any identifiability results for this particular setting, and further investigation into this matter is beyond the scope of the current work. It is interesting to note, however, that the CVAE-SCM from (13) can be understood as a non-linear extension of the linear Gaussian model with equal error variances considered by [37], for which identifiability has been shown.

In general, there seem to be very few works addressing identifiability of SCMs in the non-linear case; we refer to [38, §7.1] for an overview of existing results. Of particular interest for our setting is the post-nonlinear model of [65], which refers to the setting in which a non-linearity $g$ is applied on top of an ANM, i.e., $X_r := g_r(f_r(\mathbf{X}_{pa(r)}) + U_r)$, and for which complete conditions on $\{f_r, g_r\}$ have been provided that lead to identifiability. Given the form of the decoders $D_r$—feedforward neural networks with stacked layers of simple non-linearities applied to linear transformations of the previous layers' output—it may be possible that the CVAE-SCM from (13) can be interpreted as a nested post-nonlinear model. We consider this an interesting direction, but leave further investigations into this matter for future work.

# D Further details on CVAE training

To learn the CVAE latent variable models, we perform amortised variational inference with approximate posteriors $q$ parameterised by encoders $E_r$ in the form of neural nets with parameters $\phi_r$,

$$p_{\psi_r}(\mathbf{z}_r|x_r, \mathbf{x}_{\text{pa}(r)}) \approx q_{\phi_r}(\mathbf{z}_r|x_r, \mathbf{x}_{\text{pa}(r)}) := \mathcal{N}(\hat{\mu}_r, \hat{\sigma}_r^2), \qquad (\hat{\mu}_r, \hat{\sigma}_r^2) := E_r(x_r, \mathbf{x}_{\text{pa}(r)}; \phi_r). \qquad \text{(D.1)}$$

The training objective in form of the evidence lower bound (ELBO) given data $\{\mathbf{x}^i\}_{i=1}^n$ is given by

$$\mathcal{L}_r(\psi_r, \phi_r) = \sum_{i=1}^n \mathbb{E}_{q_{\phi_r}(\mathbf{z}|x_r^i, \mathbf{x}_{\text{pa}(r)}^i)} \left[ \left\| x_r^i - D_r(\mathbf{x}_{\text{pa}(r)}^i, \mathbf{z}; \psi_r) \right\|^2 \right] + \beta_r D_{\text{KL}} \left( q_{\phi_r}(\mathbf{z}|x_r^i, \mathbf{x}_{\text{pa}(r)}^i) \middle\| p(z) \right)$$
(D.2)

We learn both $\psi_r$ and $\phi_r$ simultaneously via stochastic gradient descend on $\mathcal{L}_r$, with gradients computed by Monte Carlo sampling from $q_{\phi_r}$ with reparametrisation. Since the pairs of encoder and decoder parameters $(\psi_r, \phi_r)$ are independent for different $r$, this can be done in parallel.

## D.1 Hyperparameter selection for CVAE training

A CVAE model was trained for every $\mathbf{X}_r|\mathbf{X}_{\text{pa}(r)}$ relation. Generally, hyperparameters were selected by comparing the distribution of real samples from the dataset against reconstructed samples from the trained CVAE obtained by sampling noise from the prior. The selection of hyperparameters was done either manually, or by performing a grid search over various encoder and decoder architectures, latent-space dimensions, and values of the hyperparameters $\beta_r$ that trade off the MSE and KL terms in the CVAE objective (D.2). For the case of automatic selection, the setup resulting in the smallest maximum mean discrepancy (MMD) statistic [14] between real and reconstructed samples was chosen as hyperparameter configuration. Further details on the search space considered and the selected values are provided in Table 5.

Table 5: Selection of hyperparameters for CVAE training was either performed manually (for Linear SCM, Non-linear ANM, Non-additve SCM) or automatically (for 7-variable semi-synthetic loan approval) by selecting the setting that resulted in the minimum MMD statistic between real and reconstructed samples.

| SCM | Conditional | Encoder Arch. | Decoder Arch. | Latent Dim. | $\lambda_{\text{KLD}}$ |
|---|---|---|---|---|---|
| Linear SCM | $X_2|X_1,$ | $1\times32\times32\times32$ | $5\times5\times1$ | 1 | 0.01 |
| | $X_3|X_1, X_2$ | $1\times32\times32\times32$ | $32\times32\times32\times1$ | 1 | 0.01 |
| Non-linear ANM | $X_2|X_1,$ | $1\times32\times32$ | $32\times32\times1$ | 5 | 0.01 |
| | $X_3|X_1, X_2$ | $1\times32\times32\times32$ | $32\times32\times1$ | 1 | 0.01 |
| Non-additve SCM | $X_2|X_1,$ | $1\times32\times32\times32$ | $32\times32\times1$ | 3 | 0.5 |
| | $X_3|X_1, X_2$ | $1\times32\times32\times32$ | $5\times5\times1$ | 3 | 0.1 |
| 7-variable semi-synthetic loan approval | any | $1\times3\times3$ $1\times5\times5$ $1\times3\times3\times3$ | $2\times1$ $2\times2\times1$ $3\times3\times1$ $5\times5\times1$ $3\times3\times3\times1$ | 1,2 | 5, 1, 0.5, 0.1, 0.05, 0.01, 0.005 |

# E  Experimental details, hyperparameter choices, and specification of SCMs

## E.1  Specification of SCMs used in our experiments

The following is a specification of all SCMs used in our experiments on synthetic and semi-synthetic data, both for data generation and to evaluate the validity of recourse actions proposed by the different approaches by computing the corresponding counterfactual in the ground-truth SCMs.

In addition, we also specify the model used to generate training labels. Note, however, that these labels are only used to train a new classifier (e.g., a logistic regression, multi-layer perceptron, or random forest) from scratch: this is the $h(\mathbf{x})$ referred to in the main paper. The label generating process is thus only used for obtaining labels to train a classifier on and is subsequently disregarded in favour of $h$.

In selecting the structural equations and label generating process, we tried to pick combinations that resulted in roughly centred features, as well as roughly balanced datasets (i.e., with a similar proportion of positive and negative training examples) that are not perfectly linearly-separable (i.e., with some class overlap). Moreover, we tried to select settings that result in a diverse set of intervention targets selected by the oracle for different factual instances, i.e., we try to avoid situations in which the optimal action is to always intervene on the same (set of) variable(s). To induce more interesting behaviour, we sample root nodes from mixtures of Gaussians.

### E.1.1  3-variable synthetic SCMs used for Table 1

A visual summary of the 3-variable synthetic SCMs used for Table 1 is provided in Fig. 4.

(a) Linear SCM        (b) Non-linear ANM        (c) Non-additive SCM

Figure 4: Histograms and scatter plots of pairwise feature relations for the synthetic 3-variable SCMs.

**Linear SCM:**  The linear 3-variable SCM consists of the following structural equations and noise distributions:

$$X_1 := U_1, \qquad\qquad\qquad U_1 \sim \mathrm{MoG}\Big(0.5\mathcal{N}(-2, 1.5) + 0.5\mathcal{N}(1, 1)\Big) \qquad \text{(E.1)}$$

$$X_2 := -X_1 + U_2, \qquad\qquad U_2 \sim \mathcal{N}(0, 1) \qquad\qquad\qquad\qquad\qquad\qquad \text{(E.2)}$$

$$X_3 := 0.05X_1 + 0.25X_2 + U_3, \qquad U_3 \sim \mathcal{N}(0, 1) \qquad\qquad\qquad\qquad\qquad\qquad \text{(E.3)}$$

**Non-linear ANM:**  The non-linear 3-variable ANM consists of the following structural equations and noise distributions:

$$X_1 := U_1, \qquad\qquad\qquad\qquad U_1 \sim \mathrm{MoG}\Big(0.5\mathcal{N}(-2, 1.5) + 0.5\mathcal{N}(1, 1)\Big) \qquad \text{(E.4)}$$

$$X_2 := -1 + \frac{3}{1 + e^{-2X_1}} + U_2, \qquad U_2 \sim \mathcal{N}(0, 0.1) \qquad\qquad\qquad\qquad\qquad\qquad \text{(E.5)}$$

$$X_3 := -0.05X_1 + 0.25X_2^2 + U_3, \qquad U_3 \sim \mathcal{N}(0, 1) \qquad\qquad\qquad\qquad\qquad\qquad \text{(E.6)}$$

**Non-additve SCM:**  The non-additive 3-variable SCM consists of the following structural equations and noise distributions:

$$X_1 := U_1, \qquad\qquad\qquad\qquad\quad U_1 \sim \mathrm{MoG}\Big(0.5\mathcal{N}(-2.5, 1) + 0.5\mathcal{N}(2.5, 1)\Big) \qquad \text{(E.7)}$$

$$X_2 := 0.25\,\mathrm{sgn}(U_2)X_1^2(1 + U_2^2), \qquad U_2 \sim \mathcal{N}(0, 0.25) \qquad\qquad\qquad\qquad\qquad \text{(E.8)}$$

$$X_3 := -1 + 0.1\,\mathrm{sgn}(U_3)(X_1^2 + X_2^2) + U_3, \qquad U_3 \sim \mathcal{N}(0, 0.25^2) \qquad\qquad\qquad \text{(E.9)}$$

**Label generation:** For all 3-variable SCMs, labels $Y$ were sampled according to

$$Y \sim \text{Bernoulli}\left(\left(1 + e^{-2.5\rho^{-1}(X_1 + X_2 + X_3)}\right)^{-1}\right) \tag{E.10}$$

where $\rho$ is the average of $(X_1 + X_2 + X_3)$ across all training samples.

### E.1.2   7-variable semi-synthetic loan approval SCM used for Table 2

For the semi-synthetic dataset, we wanted to capture some relations between the involved variables that seemed somewhat intuitive to us and to some limited extent reflect a loan approval setting in the real-world:

- loan amount and duration being largest for mid-aged people who may want to build a house and start a family, and smaller for younger and older people;
- loan duration increasing with loan amount due to the an upper limit on monthly payments that can be afforded
- savings increasing once income passes a certain (minimal-sustenance) threshold;
- income increasing with age;
- education increasing with age initially before eventually saturating;
- gender differences in income and (access to) education due to existing gender-discrimination and inequality of opportunities in the population;

A visual summary of the 7-variable semi-synthetic loan SCM is shown in Fig. 5.

**Semi-synthetic SCM:** The loan approval SCM consists of the following structural equations and noise distributions:

$$
\begin{aligned}
G &:= U_G, & U_G &\sim \text{Bernoulli}(0.5) & \text{(E.11)}\\
A &:= -35 + U_A, & U_A &\sim \text{Gamma}(10, 3.5) & \text{(E.12)}\\
E &:= -0.5 + \left(1 + e^{-\left(-1 + 0.5G + \left(1 + e^{-0.1A}\right)^{-1} + U_E\right)}\right)^{-1}, & U_E &\sim \mathcal{N}(0, 0.25) & \text{(E.13)}\\
L &:= 1 + 0.01(A - 5)(5 - A) + G + U_L, & U_L &\sim \mathcal{N}(0, 4) & \text{(E.14)}\\
D &:= -1 + 0.1A + 2G + L + U_D, & U_D &\sim \mathcal{N}(0, 9) & \text{(E.15)}\\
I &:= -4 + 0.1(A + 35) + 2G + GE + U_I, & U_I &\sim \mathcal{N}(0, 4) & \text{(E.16)}\\
S &:= -4 + 1.5\mathbb{I}_{\{I > 0\}}I + U_S, & U_S &\sim \mathcal{N}(0, 25) & \text{(E.17)}
\end{aligned}
$$

Note that variables in the above SCM often have a relative meaning in terms of deviation from the mean, e.g., we centre the Gamma-distributed age around its mean of 35, so that $A$ has the meaning of "age-difference from the mean of 35" (and similarly for other variables).

**Label generation:** Labels $Y$ were sampled according to

$$Y \sim \text{Bernoulli}\left(\left(1 + e^{-0.3(-L - D + I + S + IS)}\right)^{-1}\right). \tag{E.18}$$

Note that this label generation process only depends on loan duration and amount, income and savings, but not on gender, age or education level.

Figure 5: Histograms and scatter plots of pairwise feature relations for the semi-synthetic loan SCM.

# F Derivation of a Monte-Carlo estimator for the gradient of the variance

We now derive an estimator for the gradient of the square-root of the variance (i.e., standard deviation) of $h$ over the interventional or counterfactual distribution of $\mathbf{X}_{d(\mathcal{I})}$ w.r.t. $\boldsymbol{\theta}$, which appears (multiplied by $\lambda_{\mathrm{LCB}}$) in the threshold $\mathtt{tresh}(a)$ of the optimisation constraint/regulariser.

First, we use the chain rule of differentiation to write

$$\nabla_{\boldsymbol{\theta}}\sqrt{\mathbb{V}_{\mathbf{x}_{d(\mathcal{I})}}\left[h\left(\mathbf{X}_{d(\mathcal{I})},\boldsymbol{\theta},\mathbf{x}_{nd(\mathcal{I})}^{\mathsf{F}}\right)\right]} = \frac{\nabla_{\boldsymbol{\theta}}\mathbb{V}_{\mathbf{x}_{d(\mathcal{I})}}\left[h\left(\mathbf{X}_{d(\mathcal{I})},\boldsymbol{\theta},\mathbf{x}_{nd(\mathcal{I})}^{\mathsf{F}}\right)\right]}{2\sqrt{\mathbb{V}_{\mathbf{x}_{d(\mathcal{I})}}\left[h\left(\mathbf{X}_{d(\mathcal{I})},\boldsymbol{\theta},\mathbf{x}_{nd(\mathcal{I})}^{\mathsf{F}}\right)\right]}} \tag{F.1}$$

Next, we write the variance as expectation and—assuming the interventional or counterfactual distribution of $\mathbf{X}_{d(\mathcal{I})}$ admits reparametrisation as is the case for the GP-SCM and CVAE models used in this paper—use the reparametrisation trick to differentiate through the expectation operator as in (15).

$$\nabla_{\boldsymbol{\theta}}\mathbb{V}_{\mathbf{x}_{d(\mathcal{I})}}\left[h\left(\mathbf{X}_{d(\mathcal{I})},\boldsymbol{\theta},\mathbf{x}_{nd(\mathcal{I})}^{\mathsf{F}}\right)\right] \tag{F.2}$$

$$= \nabla_{\boldsymbol{\theta}}\mathbb{E}_{\mathbf{X}_{d(\mathcal{I})}}\left[\left(h\left(\mathbf{X}_{d(\mathcal{I})},\boldsymbol{\theta},\mathbf{x}_{nd(\mathcal{I})}^{\mathsf{F}}\right) - \mathbb{E}_{\mathbf{X}_{d(\mathcal{I})}'}\left[h\left(\mathbf{X}_{d(\mathcal{I})}',\boldsymbol{\theta},\mathbf{x}_{nd(\mathcal{I})}^{\mathsf{F}}\right)\right]\right)^2\right] \tag{F.3}$$

$$= \nabla_{\boldsymbol{\theta}}\mathbb{E}_{\mathbf{z}\sim\mathcal{N}(\mathbf{0},\mathbf{I})}\left[\left(h\left(\mathbf{X}_{d(\mathcal{I})}(\mathbf{z};\boldsymbol{\theta}),\boldsymbol{\theta},\mathbf{x}_{nd(\mathcal{I})}^{\mathsf{F}}\right) - \mathbb{E}_{\mathbf{z}'\sim\mathcal{N}(\mathbf{0},\mathbf{I})}\left[h\left(\mathbf{x}_{d(\mathcal{I})}(\mathbf{z}';\boldsymbol{\theta}),\boldsymbol{\theta},\mathbf{x}_{nd(\mathcal{I})}^{\mathsf{F}}\right)\right]\right)^2\right] \tag{F.4}$$

$$= \mathbb{E}_{\mathbf{z}\sim\mathcal{N}(\mathbf{0},\mathbf{I})}\left[\nabla_{\boldsymbol{\theta}}\left(h\left(\mathbf{X}_{d(\mathcal{I})}(\mathbf{z};\boldsymbol{\theta}),\boldsymbol{\theta},\mathbf{x}_{nd(\mathcal{I})}^{\mathsf{F}}\right) - \mathbb{E}_{\mathbf{z}'\sim\mathcal{N}(\mathbf{0},\mathbf{I})}\left[h\left(\mathbf{x}_{d(\mathcal{I})}(\mathbf{z}';\boldsymbol{\theta}),\boldsymbol{\theta},\mathbf{x}_{nd(\mathcal{I})}^{\mathsf{F}}\right)\right]\right)^2\right] \tag{F.5}$$

$$= \mathbb{E}_{\mathbf{z}\sim\mathcal{N}(\mathbf{0},\mathbf{I})}\left[2\left(h\left(\mathbf{X}_{d(\mathcal{I})}(\mathbf{z};\boldsymbol{\theta}),\boldsymbol{\theta},\mathbf{x}_{nd(\mathcal{I})}^{\mathsf{F}}\right) - \mathbb{E}_{\mathbf{z}'\sim\mathcal{N}(\mathbf{0},\mathbf{I})}\left[h\left(\mathbf{x}_{d(\mathcal{I})}(\mathbf{z}';\boldsymbol{\theta}),\boldsymbol{\theta},\mathbf{x}_{nd(\mathcal{I})}^{\mathsf{F}}\right)\right]\right) \tag{F.6}$$

$$\times \left(\nabla_{\boldsymbol{\theta}}h\left(\mathbf{X}_{d(\mathcal{I})}(\mathbf{z};\boldsymbol{\theta}),\boldsymbol{\theta},\mathbf{x}_{nd(\mathcal{I})}^{\mathsf{F}}\right) - \mathbb{E}_{\mathbf{z}'\sim\mathcal{N}(\mathbf{0},\mathbf{I})}\left[\nabla_{\boldsymbol{\theta}}h\left(\mathbf{x}_{d(\mathcal{I})}(\mathbf{z}';\boldsymbol{\theta}),\boldsymbol{\theta},\mathbf{x}_{nd(\mathcal{I})}^{\mathsf{F}}\right)\right]\right)\right] \tag{F.7}$$

We can now obtain an estimate of the gradient with two independent sets of Monte Carlo samples of $\mathbf{X}_{d(\mathcal{I})}$, drawn via reparametrisation from the interventional or counterfactual distribution,

$$\{\mathbf{x}_{d(\mathcal{I})}^{(m)} := \mathbf{x}_{d(\mathcal{I})}(\mathbf{z}^{(m)};\boldsymbol{\theta})\}_{m=1}^{M}, \quad \{\mathbf{x}_{d(\mathcal{I})}^{(m')} := \mathbf{x}_{d(\mathcal{I})}(\mathbf{z}^{(m')};\boldsymbol{\theta})\}_{m'=1}^{M'} \quad \text{where} \quad \mathbf{z}^{(m)},\mathbf{z}^{(m')} \overset{\text{i.i.d.}}{\sim} \mathcal{N}(\mathbf{0},\mathbf{I}). \tag{F.8}$$

This yields the following Monte Carlo gradient estimator of the variance:

$$\nabla_{\boldsymbol{\theta}}\mathbb{V}_{\mathbf{x}_{d(\mathcal{I})}}\left[h\left(\mathbf{X}_{d(\mathcal{I})},\boldsymbol{\theta},\mathbf{x}_{nd(\mathcal{I})}^{\mathsf{F}}\right)\right] \approx \frac{1}{M}\sum_{m=1}^{M}\left[2\left(h\left(\mathbf{x}_{d(\mathcal{I})}^{(m)},\boldsymbol{\theta},\mathbf{x}_{nd(\mathcal{I})}^{\mathsf{F}}\right) - \frac{1}{M'}\sum_{m'=1}^{M}h\left(\mathbf{x}_{d(\mathcal{I})}^{(m')},\boldsymbol{\theta},\mathbf{x}_{nd(\mathcal{I})}^{\mathsf{F}}\right)\right) \tag{F.9}$$

$$\times \left(\nabla_{\boldsymbol{\theta}}h\left(\mathbf{x}_{d(\mathcal{I})}^{(m)},\boldsymbol{\theta},\mathbf{x}_{nd(\mathcal{I})}^{\mathsf{F}}\right) - \frac{1}{M'}\sum_{m'=1}^{M'}\nabla_{\boldsymbol{\theta}}h\left(\mathbf{x}_{d(\mathcal{I})}^{(m')},\boldsymbol{\theta},\mathbf{x}_{nd(\mathcal{I})}^{\mathsf{F}}\right)\right)\right] \tag{F.10}$$

Substituting the above expression, together with the following Monte Carlo estimate of the (undifferentiated) variance

$$\mathbb{V}_{\mathbf{x}_{d(\mathcal{I})}}\left[h\left(\mathbf{X}_{d(\mathcal{I})},\boldsymbol{\theta},\mathbf{x}_{nd(\mathcal{I})}^{\mathsf{F}}\right)\right] \approx \frac{1}{M-1}\sum_{m=1}^{M}\left(h\left(\mathbf{x}_{d(\mathcal{I})}^{(m)},\boldsymbol{\theta},\mathbf{x}_{nd(\mathcal{I})}^{\mathsf{F}}\right) - \frac{1}{M}\sum_{m'=1}^{M'}h\left(\mathbf{x}_{d(\mathcal{I})}^{(m')},\boldsymbol{\theta},\mathbf{x}_{nd(\mathcal{I})}^{\mathsf{F}}\right)\right)^2, \tag{F.11}$$

into (F.1) gives the desired estimate for the gradient of the standard deviation of $h$.