[Reviews · NeurIPS 2020]

Review 1

Summary and Contributions: This paper proposes a new method for algorithmic recourse when complete causal knowledge maybe unavailable. Under the assumption that the causal graph is known (but not the structural equations), i) A negative result is proved suggesting that without knowing structural equations, recourse cannot be guaranteed. ii) For the class of structural equations with additive gaussian noise, recourse is framed using a counterfactual query to the SCM along with an algrorithm that can help estimate the counterfactual distribution (while accounting for uncertainty over the functional form of the SCM). iii) Additive gaussian noise assumptions are further relaxed where recourse is now formulated as an interventional query that conditions on a subpopulation (determined by similarity over a subset of features determined by the causal graph) under the different interventions (potential feasible actions that could change outcomes). Finally experimental evaluation demonstrates the benefits of each of these methods on extensive evaluation.

Strengths: The paper is well motivated, results are technically as well as practically interesting. This is in my opinion a significant contribution the area of actionable recourse in fairness literature and is definitely relevant to the NeurIPS audience. The results of GP-SCMs are probably of independent interest and authors could highlight that more strongly.

Weaknesses: 1. High level comment: Since the proposed solutions are both based on a counterfactual as well as interventional query, it seems important to justify which framing is more appropriate for the recourse task. On the face of it, recourse is a counterfactual query rather than an interventional one. The justification and motivation for each potential formulation is unclear and it would be good if the authors incorporate that in their motivation. 2. Authors do not address the issue of feasibility in the level of detail as is warranted for recourse. This makes the formulation a little impractical for actual practice. Can the authors clarify the details of feasibility or enumeration of feasible actions? All comments in the paper allude to searching over potential intervention sets \mathcal{I}. However, there are causal dependencies in the graph which can determine allowable feasible sets. It is unclear how to address this challenge here. 4. Is the learned CVAE completely respect the causal graph? The procedure of training CVAEs for interventional recourse should be clarified in further detail. 5. In experimental evaluation, I did not see a comparison to existing baselines in recourse, neither a qualitative assessment of the type of recourses that the model learns.

Correctness: I have gone over the proofs and they are reasonably detailed and correct. Empirical claims need further evaluation.

Clarity: The paper is well written, motivated and structured well. Experimental evaluation is also extensively presented.

Relation to Prior Work: The paper is clearly situated in existing literature. However, the authors do fail to cite an important critique of recourse work - The philosophical basis of algorithmic recourse by Venkatasubramanian and Alfano (FAccT* 2020) and contextualize their contributions in terms of issues raised in this work. This is important for calibrating broader impact.

Reproducibility: Yes

Additional Feedback: Edit: I have read the rebuttal, other reviews and acknowledge that authors addressed my concerns.


Review 2

Summary and Contributions: This paper provides a probabilistically correct method of recourse under a more realistic expectation of knowledge of the underlying causal structure. The paper demonstrates concretely the weakness of the most similar result, showing that there is no guaranteed recourse for unknown structural equations. Then the paper then goes on to provide two forms of recourse, one individualized and one subgroupwise; both based on GP- Structural Causal Models. Finally they proved an optimization technique to solve the proposed problems for recourse and present experimental results showing improvements in validity and cost.

Strengths: this work provides a sound solution to an important problem and extends the literature in a novel way. Providing recourse recommendations is required or desirable in many domains, so reliable techniques for doing so under realistically specified problems is important. This is relevant to the NeurIPS community as explanation is by regulation and actionable recourse solves the underlying problem in a more robust way for impacted individuals. the claims are sound and the empirical evidence is clear, enough description and discussion of propositions is included in the paper that they're beleivable, even though all proofs are left to the supplemental materials.

Weaknesses: The results are only on synthetic data, though this is understandable because true SCMs are not generally available. Collaboration with a

Correctness: the claims are well supported and correct. The method is correct at the modeling level and the algorithmic solution makes sense as a solution. The baselins of the experiments and the metrics are well described.

Clarity: Overall the paper is well written and clear. There are some parts that are difficult to parse or places where clarity could be improved. In the early sections, intervention and action seem to be used to refer to related, but different concepts, but later they seem to be used interchangeably. It could make the overall exposition clearer to use intervention for the more general causal intervention and action to the resourse- specific action. UPDATE: thanks for recognizing this and agreeing to update it Minor: Figure 1 caption is hard to parse quickly because the subpart labels are in the sentence and mixed between before and after the content that goes with them. line 119: the "not" in the dashed clause reads like a double negative, and could be removed. Figure 2c is labeled as a tradeoff between two variables, but the plots are each one versus the control parameter. Also, validity and cost increase in the same direction, because high cost is bad, it is a tradeoff in a sense, but the figure doesn't match expectations when labeled "tradeoff" line 189: put commas around nd(I) Table 2's caption incorrectly says that it si 3 variables, when the paper text says that table 2 is the 7 variable model results.

Relation to Prior Work: The work is clearly positioned in the broader context and the innovations are clearly stated.

Reproducibility: Yes

Additional Feedback:


Review 3

Summary and Contributions: The paper studies the problem of algorithmic recourse when the true underlying structural model is unknown. The paper first shows that algorithmic recourse cannot be guaranteed if the true structural equations are unknown. Then, the paper proposes two probabilistic approaches to solve the problem: 1) the first approach assumes the structural equations to be additive noise models (ANMs) with Gaussian noise and estimates a counterfactual distribution under this assumption, 2) the second approach estimates the effect of interventions on a subpopulation around the given factual datapoint. All approaches have been evaluated on three classes of SCMs for a synthetic 3-variable case and a semi-synthetic 7-variable loan approval problem.

Strengths: - The paper builds upon the idea of counterfactual explanation to generate actionable recommendations that leads to a change in prediction. - The paper does not treat features as independently manipulable and takes the causal structure between features into account. - The proposed approaches don’t need the true underlying structural model to work. - The paper is written very well.

Weaknesses: - What are the computation costs for the proposed approaches and how do they compare to non-probabilistic baselines? - For 3-variable case, non-probabilistic baselines are performing as well as CATE-CVAE for non-additive SCM and kernel regression baseline is performing as well as CATE-CVAE for non-linear SCM. - In both experiments, the subpopulation-based approach CATE-CVAE is underperforming in terms of validity as well as cost compared to the pseudo-counterfactual M-CVAE. The paper should include a scenario/case where subpopulation-based approach might be preferable. - The authors should include some application-based examples in the paper to highlight the advantages of their approach as compared to prior works as being done in [7]. ------------------------------------------------------------------------------------------------------------------------------------------------------------------------------------------------ Update after rebuttal: I thank the authors for their responses to my questions/concerns. They satisfactorily answer most of my concerns. Therefore, I am upgrading my rating.

Correctness: The work is technically sound. I have some questions/concerns about the experimental results, as highlighted in the weaknesses section above.

Clarity: The paper is very well written.

Relation to Prior Work: Yes, the paper situates itself well among prior works.

Reproducibility: Yes

Additional Feedback:

[Author Response · NeurIPS 2020]

Thank you for the thoughtful, detailed, and overall very positive feedback. Reviewers agree that the paper is *well written* (**R1**, **R2**, **R3**), *well motivated* (**R1**), *clear* (**R2**), and *novel* (**R1**, **R2**). They consider the topic *important* (**R2**) and our methodology *sound* (**R2**, **R3**), *technically as well as practically interesting*, and *of independent interest* (**R1**). Reviewers find *the experimental evaluation extensive* (**R1**) and *well described* (**R2**), *our empirical evidence clear*, and *claims well supported and correct* (**R2**). We address the main concerns below and will incorporate all reviewers' feedback.

**R1** (+**R3**): *Recourse is a counterfactual query, but the proposed solutions are based on counterfactual as well as interventional queries. Which framing is more appropriate?* We mostly agree that recourse is predominantly a counterfactual query. This is certainly the case in static environments where any changes result from an individual's actions. In dynamic settings, however, the unobserved SCM background variables $U_r$ can also capture environmental factors which may change, e.g., between consecutive loan applications, and should thus not be kept fixed as in the counterfactual approach. Depending on the setting, recourse may thus constitute a mixture of interventional and counterfactual queries. Independently of this fundamental question, our interventional notion of recourse is practically motivated by a reluctance to make (parametric) assumptions about the structural equations which are both untestable but also necessary for counterfactual reasoning (see also Example 1; l.118–120; 181–186). In such cases, the subpopulation-based approach can provide a useful alternative. In analogy, in epidemiology one ideally wants to estimate the individualised (i.e., counterfactual) treatment effect, but (without additional assumptions) has to resort to the (conditional) average (i.e., interventional) effect instead. Finally, we note that the proposed approaches can be combined depending on the available domain knowledge since each causal relation is treated separately.

**R1**, **R3**: *Lack of comparison to existing baselines and qualitative assessment of the type of recourse that the model learns.* We compare with an oracle (known SCM) corresponding to [7], as well as with point-based linear and kernel-ridge regression baselines. A comparison with approaches assuming independent features raises issues such as whether descendant variables are allowed to change or whether keeping their value constant should incur a cost (and if so how much), and are thus neither natural nor straight-forward. We therefore restrict ourselves to a comparison with *causal* recourse approaches. For a qualitative assessment of the recommended actions, we refer to Table 5 in Appendix E.

**R1**: *Does the learned* CVAE *respect the causal graph? The procedure of training* CVAEs *for interventional recourse should be clarified.* Yes, the interventional approaches respect the causal graph: we do not train a single CVAE model, but—motivated by the identifiability result (11) in Proposition 4—instead learn a separate CVAE/GP for each parent-child conditional $p(X_r|\mathbf{X}_{\text{pa}(r)})$, see also l.193–201. We apologise for failing to convey this important point.

**R1**: *Can the authors clarify the enumeration of feasible actions? There are causal dependencies in the graph which can determine allowable feasible sets.* We agree that feasibility is an essential aspect of recourse. However, our understanding is that causal dependencies in the graph *do not* determine allowable feasible sets. E.g., consider a mutable but non-actionable variable BMI (body mass index), a descendant of an actionable variable weight and an immutable variable gender. The mutability and actionability constraints above are determined by context-/individual-specific constraints in the application domain (summarised in $\mathbb{A}^{\text{F}}$), rather than by the mathematical framework of SCMs.

**R1**: *The authors do fail to cite an important critique of recourse work [VA20][1] and contextualize their contributions in terms of issues raised in this work.* Thank you for the pointer. Indeed, concerns as the ones raised in [VA20] were a primary motivation for our paper. Specifically, [VA20] explore a number of challenges facing algorithmic recourse, and argue (alongside [BSR20][2]) that we need to look beyond independently manipulable features; [7] then phrases this as a *causal* problem within the SCM framework. We apologize for failing to cite [VA20, BSR20] and will correct this.

**R2**: *It is not clear whether "action" and "intervention" refer to slightly different concepts or not.* Thank you for pointing this out; we will revise the wording to clarify that we consider actions and interventions as interchangeable.

**R3**: *How do computational complexity of probabilistic and point-based approaches compare?* Their main difference is the expectation in the constraint whose approximation scales linearly in the number of Monte Carlo samples.

**R3**: *Some parts of the results ($\mathcal{M}_{\text{KR}}$ on nonlinear* ANM, CATE$_{\text{CVAE}}$ *vs* $\mathcal{M}_{\text{CVAE}}$, *non-additive SCM) are not convincing. The paper should include a scenario where subpopulation-based approaches are preferable.* The strong performance of $\mathcal{M}_{\text{KR}}$ on the nonlinear ANM is expected as its assumptions are satisfied. Moreover, CATE$_{\text{CVAE}}$ indeed outperforms $\mathcal{M}_{\text{CVAE}}$ on the same. We agree regarding the non-additive SCM (where subpopulation-based approaches should be preferred) and (i) found a more convincing example by also making $X_3|X_1, X_2$ multimodal. In general, misspecified models tend to intervene on leaf nodes (due to failure to capture causal relations) and thereby can still achieve recourse, though at higher cost. We also tried (ii) making leaf nodes non-actionable (such as BMI from the example above) and (iii) reducing the cost of acting on root nodes, both of which also make the advantageous cost-validity trade-off of our methods more pronounced. We will include these additional findings in the revised version.

Overall, we hope that the above will convince all reviewers that *"this is a significant contribution [in] the area of actionable recourse"* (**R1**) and that it *"is definitely relevant to the NeurIPS audience"* (**R1**, **R2**).

## Footnotes

[1] [VA20] Venkatasubramanian, Alfano. "The philosophical basis of algorithmic recourse." FAccT 2020.

[2] [BSR20] Barocas, Selbst, Raghavan. "The hidden assumptions behind counterfactual explanations and principal reasons." FAccT 2020.


[Meta-Review · NeurIPS 2020]

This paper proposes a new method for algorithmic recourse when complete causal knowledge may be unavailable. The paper is well motivated, is of practical interest, and is technically sound. It also makes a very interesting and strong contribution in the space of algorithmic recourse and therefore we recommend an acceptance. We would strongly encourage the authors to address the following points in the rebuttal: 1. Please compare the proposed algorithm to prior work by Mahajan et. al. 2019 and/or other recourse generation algorithms. 2. Results are currently show only on synthetic data. Please try and incorporate at least one experiment with a real world dataset.